# The small acid-soluble proteins of *Clostridioides difficile* regulate sporulation in a SpoIVB2-dependent manner

**Hailee N. Nerber, Marko Baloh, Joshua N. Brehm, Joseph A. Sorg** *

Department of Biology, Texas A&M University, College Station, Texas, United States of America

* jsorg@bio.tamu.edu

## Abstract

*Clostridioides difficile* is a pathogen whose transmission relies on the formation of dormant endospores. Spores are highly resilient forms of bacteria that resist environmental and chemical insults. In recent work, we found that *C. difficile* SspA and SspB, two small acid-soluble proteins (SASPs), protect spores from UV damage and, interestingly, are necessary for the formation of mature spores. Here, we build upon this finding and show that *C. difficile sspA* and *sspB* are required for the formation of the spore cortex layer. Moreover, using an EMS mutagenesis selection strategy, we identified mutations that suppressed the defect in sporulation of *C. difficile* SASP mutants. Many of these strains contained mutations in *CDR20291_0714* (*spoIVB2*) revealing a connection between the SpoIVB2 protease and the SASPs in the sporulation pathway. This work builds upon the hypothesis that the small acid-soluble proteins can regulate gene expression.

## Author summary

In prior work, we found that the *Clostridioides difficile* small acid-soluble proteins, SspA and SspB, are important for the UV resistance that is characteristic of dormant spores (as found in many other spore-forming bacteria). Surprisingly, we also found that the combinatorial deletion of *sspA* and *sspB* led to a block in the ability of *C. difficile* to form dormant spores, a phenotype not observed in other sporulating bacteria. Here, we build upon this work and find that this block can be overcome by mutations in *spoIVB2* –σ^F-regulated protease. Our data suggest that the σ^G-regulated SspA and SspB directly or indirectly activate *spoIVB2* expression during late stages of sporulation and that the identified *spoIVB2* alleles overcome the absence of *sspA* and *sspB* by increasing *spoIVB2* translation efficiency and, thus, abundance during late stages of sporulation. Our data build upon the hypothesis that small acid-soluble proteins can regulate gene expression and indicate that other spore-forming bacteria may have unidentified targets that these proteins may regulate during the sporulation process.

**Data Availability Statement:** The authors confirm that all data underlying the findings are fully available without restriction. All relevant data are

within the paper and its Supporting Information files.

**Funding:** This project was supported by the Division of Microbiology and Infectious Diseases, National Institute of Allergy and Infectious Diseases (R01AI116895 and R01AI172043 to JAS). The content is solely the responsibility of the authors and does not necessarily represent the official views of the NIAID. The funders had no role in study design, data collection and analysis, decision to publish, or preparation of the manuscript.

**Competing interests:** The authors have declared that no competing interests exist.

## Introduction

*Clostridioides difficile* is a Gram-positive pathogen that causes approximately 220,000 cases of infection and nearly 13,000 deaths annually [1]. *C. difficile* vegetative cells produce toxins that disrupt the colonic epithelium, resulting in diarrhea and colonic inflammation [2,3]. These toxin-producing vegetative cells are strictly anaerobic and cannot survive outside of a host for extended periods [4]. However, *C. difficile* produces endospores that are shed into the environment, can withstand oxygen and other environmental insults, and serve as the transmissive form of the organism [5–7].

Endospores are highly structured forms of bacteria. Residing in the spore core are the DNA, RNA, ribosomes, calcium dipicolinic acid (Ca-DPA), small acid-soluble proteins (SASPs), and other proteins that are necessary for the spore to outgrow into a vegetative cell [8–13]. Surrounding the core is a phospholipid membrane, cell wall, and a specialized cortex peptidoglycan layer. In the cortex, many of the N-acetylmuramic acid residues are converted into muramic-δ-lactam residues, which are recognized by the spore cortex lytic enzymes during germination [8,14–16]. Outside of the cortex is a phospholipid outer membrane, a proteinaceous spore coat, and an exosporium [8,17–21].

Generally, endospores are formed in response to nutrient deprivation. Upon initiation of sporulation, the vegetative cell asymmetrically divides into the larger mother cell and the smaller forespore compartments [22,23]. The forespore becomes engulfed by the mother cell so that it can be matured into the dormant endospore. Once the endospore is fully formed, the mother cell lyses and releases the spore into the environment [24].

Like all known endospore-forming bacteria, the *C. difficile* sporulation program initiates upon phosphorylation of the sporulation master transcriptional activator, Spo0A [5, 25, 26]. After asymmetric division, each compartment begins a cascade of sigma factor activation [27,28]. In the mother cell compartment, $\sigma^E$ becomes activated and leads to $\sigma^K$ expression. In the forespore compartment, $\sigma^F$ is activated and leads to $\sigma^G$ activation [8,23]. Loss of $\sigma^F$ results in a strain that does not complete engulfment or form the cortex layer [28]. The loss of $\sigma^G$ results in a strain that forms a localized coat layer but does not fully complete engulfment (i.e. no membrane fission) or form the cortex layer [27]. Loss of $\sigma^E$ results in a strain that is blocked at asymmetric septation. Loss of $\sigma^K$ results in a strain that fully engulfs the forespore and forms a correctly localized cortex layer, but no visible coat layer [28]. Thus, cortex assembly occurs through $\sigma^G$ regulated genes and coat production is dependent on $\sigma^K$ genes.

The small acid-soluble proteins (SASPs) are very abundant in spores and have high sequence similarity across spore-forming species [29]. In many organisms, including *Bacillus subtilis* and *Clostridium perfringens*, the SASPs protect DNA against UV damage and damage from genotoxic chemicals [30–33]. In *B. subtilis*, the SASPs are considered non-specific DNA binding proteins that coat the DNA and change the conformation to a more rigid, intermediate, B to A form [29,34–37]. This conformation leads to difficulty in forming UV-induced thymidine-dimers and, instead, promotes the formation of spore photoproducts; a repair mechanism is present in the spore to correct these lesions [38–40]. In *in vitro* transcription assays, addition of SASPs to DNA reduced transcription of some, but not all, genes, further illustrating their ability to bind DNA [37]. Moreover, the absence of transcription in mutant strains whose spores cannot degrade SASPs, suggest that SASPs could regulate gene expression [37,41].

In prior work, we found that the *C. difficile* SASPs are important for spore UV resistance but do not strongly contribute to chemical resistances [42]. Surprisingly, a *C. difficile* Δ*sspA* Δ*sspB* double mutant strain could not complete spore formation, a phenotype not observed in other endospore-forming bacteria. This led us to hypothesize that the *C. difficile* SASPs are

involved, somehow, in regulating sporulation. We hypothesize that SASPs have regions of high affinity on DNA where they bind to influence the transcription of genes. As the concentration of SASPs increases, they nonspecifically coat the DNA to provide the protection normally associated with SASPs. In the *C. difficile* Δ*sspA* Δ*sspB* strain, we hypothesize that sporulation is reduced due to altered gene expression of important sporulation genes.

Using a strategy that selected for the generation of mature spores from the sporulation deficient *C. difficile* Δ*sspA* Δ*sspB* strain, we identified mutations in *spoIVB2* that suppressed the mutant sporulation phenotype. SpoIVB2 is a protease that is recently characterized in *C. difficile* and the *C. difficile* Δ*spoIVB2* mutant strain has a phenotype similar to the *C. difficile* Δ*sspA* Δ*sspB* strain. Based upon the data in this manuscript, we hypothesize that the $\sigma^G$-dependent expression of the *C. difficile* SASPs activates the $\sigma^F$-dependent expression of *spoIVB2*, and that low levels of SpoIVB2 in a *C. difficile* Δ*sspA* Δ*sspB* mutant halts sporulation by an unknown mechanism.

## Results

### C. difficile sspA and sspB regulate sporulation in the C. difficile CD630Δerm strain

In prior work, we discovered that *C. difficile* SspA and SspB were, individually, important for UV resistance [42]. Surprisingly, we found that the combinatorial deletion of the *sspA* and *sspB* genes, or a deletion in *sspB* and an *sspA*$_{G52V}$ missense mutation (referred to as *C. difficile* Δ*sspB*\* hereafter), in the *C. difficile* R20291 strain resulted in the drastic reduction of mature spore formation and, instead, resulted in phase gray spores [42]. To confirm that this phenotype was strain independent, we generated the single and double mutants of *sspA* and *sspB* in the *C. difficile* CD630Δ*erm* strain. Unsurprisingly, the CD630Δ*erm* Δ*sspA* Δ*sspB* double mutant also produced phase gray spores that were trapped within mother cells (Fig 1A). Though the single mutants did not affect spore yield, the double mutant had a 5-$\log_{10}$ decrease in spore formation. This defect could be restored to near wildtype levels by expression of *sspA* and *sspB*, *in trans*, from a plasmid (Fig 1B). When UV resistance was assessed, the *C. difficile* CD630Δ*erm* Δ*sspA* and the Δ*sspB* single mutant strains both had an approximate 1-$\log_{10}$ loss in viability after 10 minutes of UV exposure (Fig 1C). Though consistent with the findings we observed for the *C. difficile* R20291 Δ*sspB* mutant strain, the impact on viability for the *sspA* mutant was less in the *C. difficile* CD630Δ*erm* background than in *C. difficile* R20291 background. SspA and SspB appear to regulate sporulation in both the *C. difficile* R20291 and CD630Δ*erm* strains, hence, this is likely a conserved function in *C. difficile*.

### B. subtilis sspA complements UV and sporulation phenotypes of C. difficile R20291 mutants

Due to the high sequence similarity of SASPs, and their ability to cross-complement in other organisms, we assessed whether *sspA* from *B. subtilis* would complement the phenotypes observed in the *C. difficile* SASP mutants [29,43]. Sporulation was complemented to varying degrees by the expression of *B. subtilis sspA* from the *C. difficile sspA* promoter in *C. difficile* Δ*sspB*\* and *C. difficile* Δ*sspA* Δ*sspB*. In *C. difficile* Δ*sspB*\*, expression of *B. subtilis sspA* increased spore yield by approximately 10-fold. In the *C. difficile* Δ*sspA* Δ*sspB* strain, expression of *B. subtilis sspA* increased spore yield by approximately 100-fold (Fig 2A).

Spores derived from a *C. difficile* Δ*sspA* mutant strain with a plasmid expressing *B. subtilis sspA*, under the *C. difficile sspA* native promoter, were exposed to UV light for 10 minutes and their viability assessed. *B. subtilis sspA* could partially restore UV resistance to the *C. difficile*

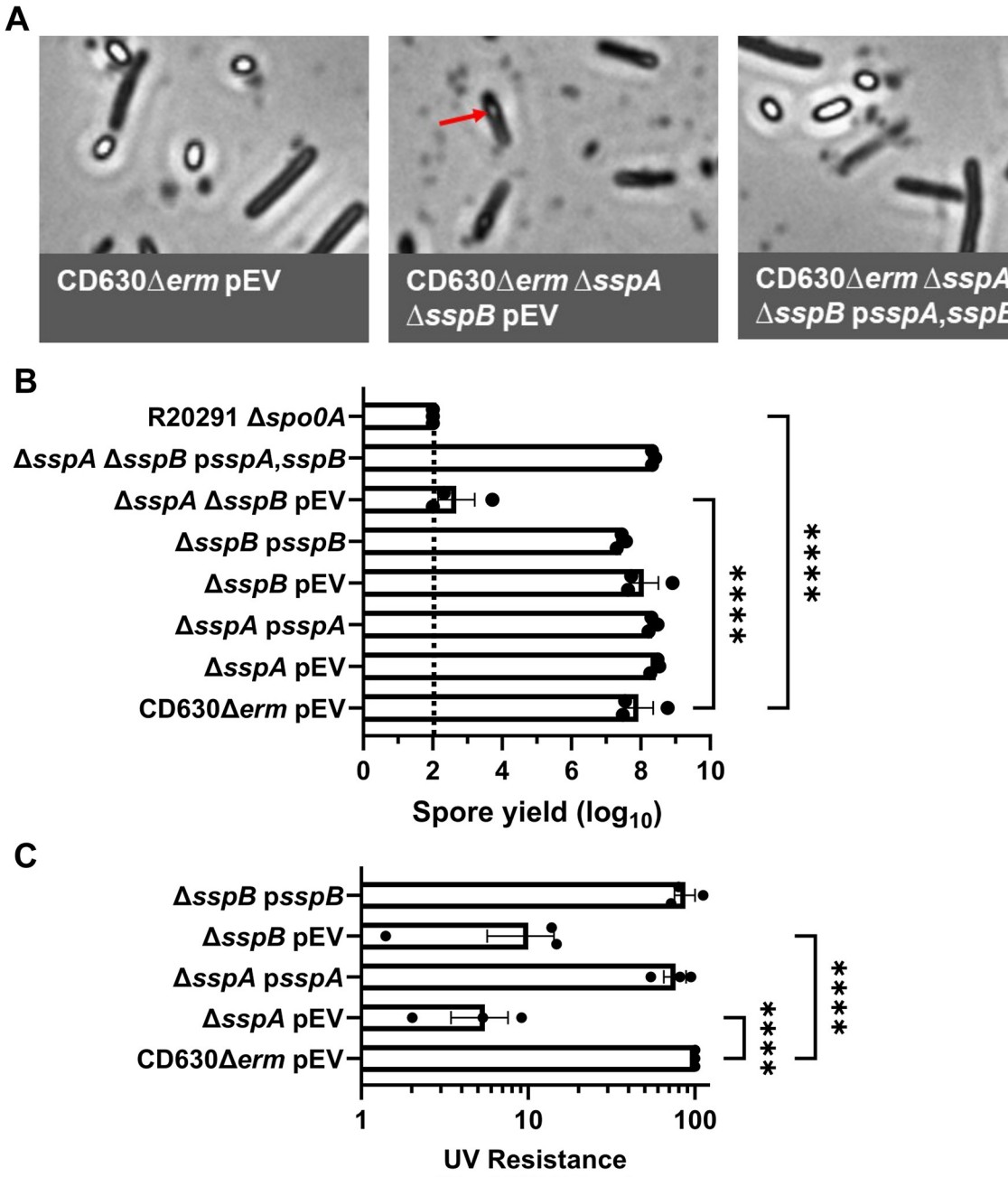

**Fig 1. Impact of *C. difficile* CD630Δ*erm sspA* and *sspB* mutations on sporulation and UV resistance.** A) Day 6 sporulating cultures were fixed in 4% formaldehyde and 2% glutaraldehyde in PBS and imaged on a Leica DM6B microscope. The red arrow represents an immature spore. B) Strains were grown on 70:30 sporulation medium for 48 hours and the cultures then were treated with 30% ethanol and plated onto rich medium supplemented with TA to enumerate spores. Spore yield was calculated by $\log_{10}$ transformation of the CFUs derived from spores. C) Spores were exposed to UV for 10 minutes with constant agitation. After treatment, they were serially diluted and plated onto rich medium supplemented with TA. The ratio of treated to untreated CFUs of the mutant strains was then compared to the ratio from WT. pEV indicates an empty plasmid within the strain. All data points represent the average of three independent experiments. Statistical analysis by one-way ANOVA with Šídák's multiple comparisons test. * $P<0.05$, ** $P<0.01$, *** $P<0.001$, **** $P<0.0001$.

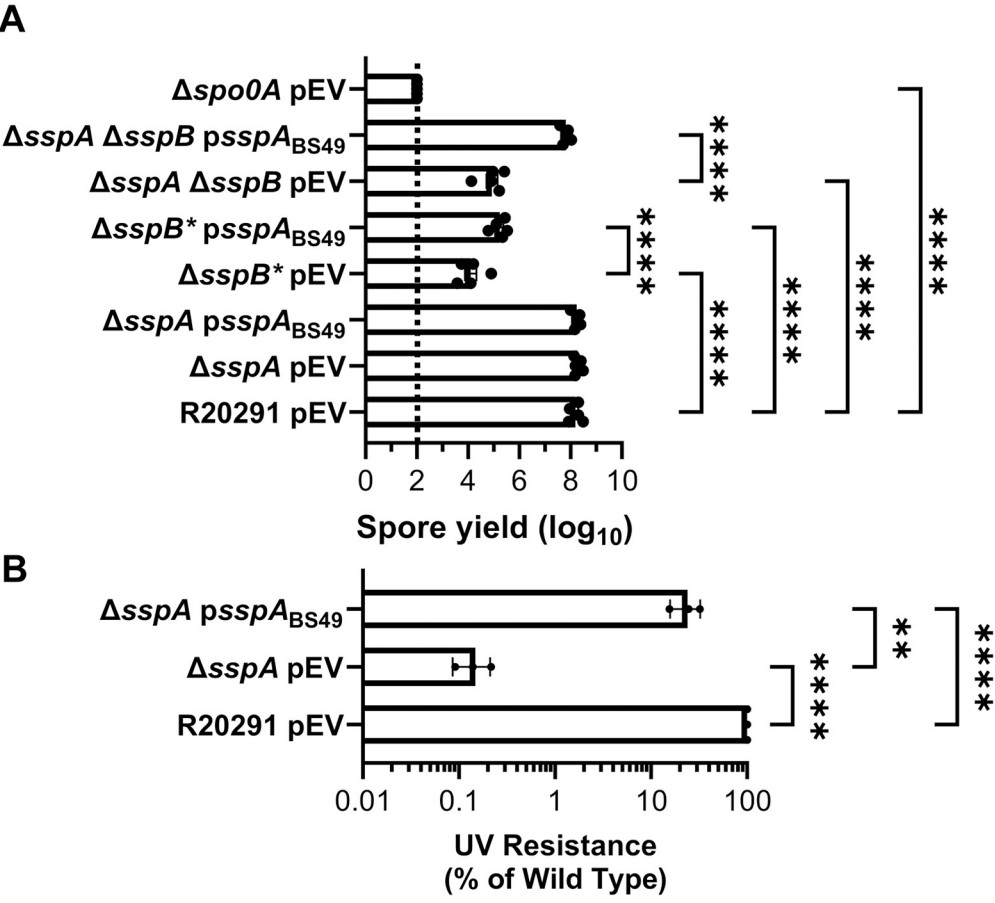

**Fig 2. *B. subtilis sspA* complements *C. difficile* SASP mutant sporulation and UV defects.** A) Spore yield of the indicated strain was determined as described in Fig 1. B) Spores were exposed to UV for 10 minutes with constant agitation. After treatment, they were serially diluted and plated onto rich medium supplemented with TA. The ratio of treated to untreated CFUs of the mutant strains was then compared to the ratio from WT. pEV indicates an empty plasmid within the strain. All data points represent the average of three independent experiments. Statistical analysis by two-way ANOVA with Šídák's multiple comparisons test. * $P<0.05$, ** $P<0.01$, *** $P<0.001$, **** $P<0.0001$.

Δ*sspA* mutant strain, although not to wild type levels (Fig 2B). These data show that *B. subtilis* and *C. difficile* SspA could function in similar ways due to the ability of *B. subtilis sspA* to complement phenotypes found in *C. difficile* SASP mutants.

## Visualizing the impact of SASP mutations on C. difficile spores

To visualize the impact of the *C. difficile* Δ*sspA* Δ*sspB* deletions on spore structure, we used transmission electron microscopy (TEM). Strains generated in the *C. difficile* R20291 background were cultured for 6 days and then prepared for TEM. As expected, the *C. difficile* R20291 wild type strain generated fully formed and mature spores. The *C. difficile* Δ*sspA* and *C. difficile* Δ*sspB* single mutant strains also formed spores with the expected spore structures (*e.g.*, cortex and coat layers). However, *C. difficile* Δ*sspB** and *C. difficile* Δ*sspA* Δ*sspB* strains generated spores that did not form cortex layers, and had a visible, but anomalous, coat and exosporium layers (Fig 3). Expression of the SASPs *in trans* under their native promoter regions complemented the mutant phenotypes by restoring formation of the cortex layer.

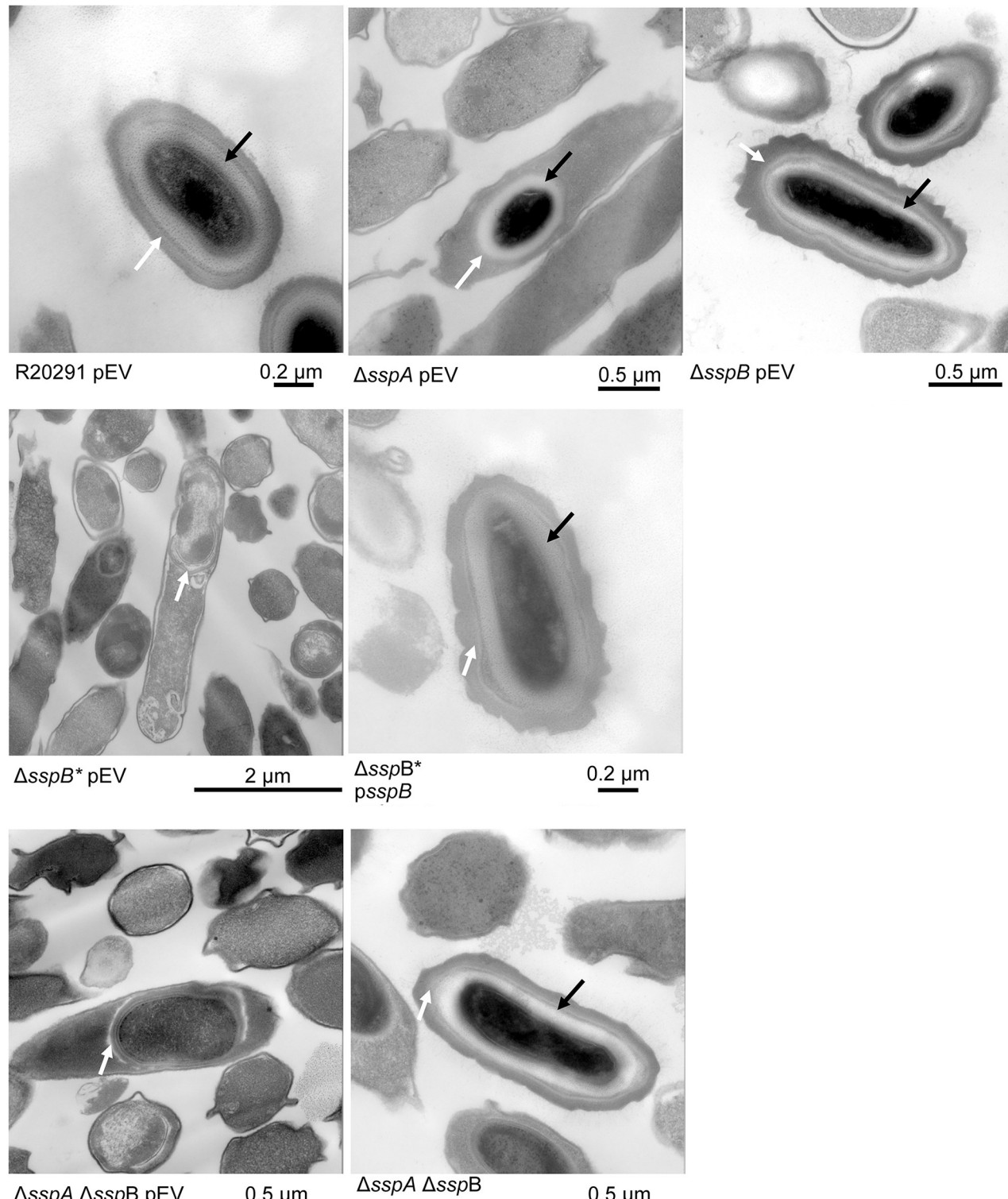

**Fig 3. SASP mutants do not form the cortex layer.** Day 6 sporulating cultures of wild type and mutant strains containing an empty vector (pEV) or the indicated plasmids were prepared for TEM. The coat layer is indicated with a white arrow, while the cortex layer is indicated with a black arrow.

## Isolating suppressor mutations of the SASP mutant phenotypes

To gain insight into how the SASPs are involved in spore formation, we used ethylmethane sulphonate (EMS) to introduce random mutations into the *C. difficile* genome, as we have previously done (S1 Fig) [44,45]. The *C. difficile* Δ*sspB*\* or the *C. difficile* Δ*sspA* Δ*sspB* strains were treated with EMS, washed, and then incubated for 5 days to generate potential spores. Subsequently, the samples were heat-treated to kill vegetative cells and immature spores. After removing cellular debris, the cultures were plated on a medium supplemented with germinant [46]. Afterwards, we isolated strains and confirmed that they generated spores. After confirmation, gDNA was extracted and sequenced to reveal the location of mutations. From independent EMS mutageneses, we identified 4 suppressor strains generated from the *C. difficile* Δ*sspB*\* strain and 11 from the *C. difficile* Δ*sspA* Δ*sspB* strain. Unsurprisingly, there were many mutations in each strain, but mutations that potentially contributed to suppression of the phenotype are listed in Table 1 (a full list of mutations can be found in S3 Table). As expected, due to the strong selection for spore dormancy, 2 out of 4 of the isolates from *C. difficile* Δ*sspB*\* had a reversion mutation in *sspA*. We identified mutations in different RNA polymerase subunits in 6 of 15 strains. These mutations could potentially affect transcription rates of various genes. Mutations within the *sigG* and *spoVT* genes were also present in some strains. *sigG* and *spoVT* mutants have a similar phenotype to the *C. difficile* Δ*sspA* Δ*sspB* strain [27,28,47,48]. Interestingly, 7 out of 15 isolates (from separate mutagenesis experiments) contained mutations in *CDR20291_0714*. Among these strains, we observed one strain with an A20T missense mutation and six with a synonymous mutation (F37F). The *C. difficile* CD630Δ*erm* genome encodes a gene homologous to *CDR20291_0714* and is annotated as *spoIVB2*. SpoIVB2 is a paralog of the SpoIVB protease, and we refer to CDR20291_0714 as SpoIVB2 from here on.

We first tested if *in trans* expression of the identified *spoIVB2* alleles could restore sporulation to the SASP mutant by generating merodiploid strains. When wild type *spoIVB2* was expressed in *C. difficile* R20291 or *C. difficile* Δ*sspB*\* the spore yield did not change from their respective phenotypes while the spore yield in the *C. difficile* Δ*sspA* Δ*sspB* strain increased by 1-$\log_{10}$ (Fig 4A). We also tested if catalytic activity impacted restoration. The catalytic site was identified by aligning *C. difficile* SpoIVB / SpoIVB2 to *B. subtilis* SpoIVB. The three catalytic residues found in *B. subtilis* are conserved in both SpoIVB and SpoIVB2 of *C. difficile* and we have used *spoIVB2*$_{S301A}$ as a catalytically dead mutant [49,50]. In the wildtype *C. difficile* R20291 strain, the spore yield was not impacted when the *spoIVB2*$_{A20T}$ or *spoIVB2*$_{F37F}$ alleles were combined with S301A (Fig 4B).

**Table 1. Shortened list of potential suppressor mutations: The mutations that could potentially suppress the sporulation defect in EMS treated isolates.** The underlined strains were derived from the *C. difficile* Δ*sspB*\* strain while the nonunderlined strains were derived from the *C. difficile* Δ*sspA* Δ*sspB* strain.

| Gene | Function | Isolates | Mutation |
|---|---|---|---|
| *rpoB* | RNA polymerase beta subunit | HNN37 | P893S |
| *rpoC* | RNA polymerase beta' subunit | HNN40 | H94Y |
| *rpoC* | RNA polymerase beta' subunit | HNN32 | W225 stp |
| *rpoC* | RNA polymerase beta' subunit | HNN40, HNN22 | S315F |
| *rpoA* | RNA polymerase alpha subunit | HNN51, HNN19 | S281F |
| *CDR20291_0714* | Stage IV sporulation protein | HNN19 | A20T |
| *CDR20291_0714* | Stage IV sporulation protein | HNN33, HNN35, HNN37, HNN38, HNN41, HNN48 | F37F; Synonymous |
| *sigG* | Forespore sporulation sigma factor | HNN39 | M97I |
| *sspA* | Small acid soluble protein | HNN26, HNN28 | V52G; Reversion |
| *spoVT* | Stage V sporulation protein | HNN51 | P39S |

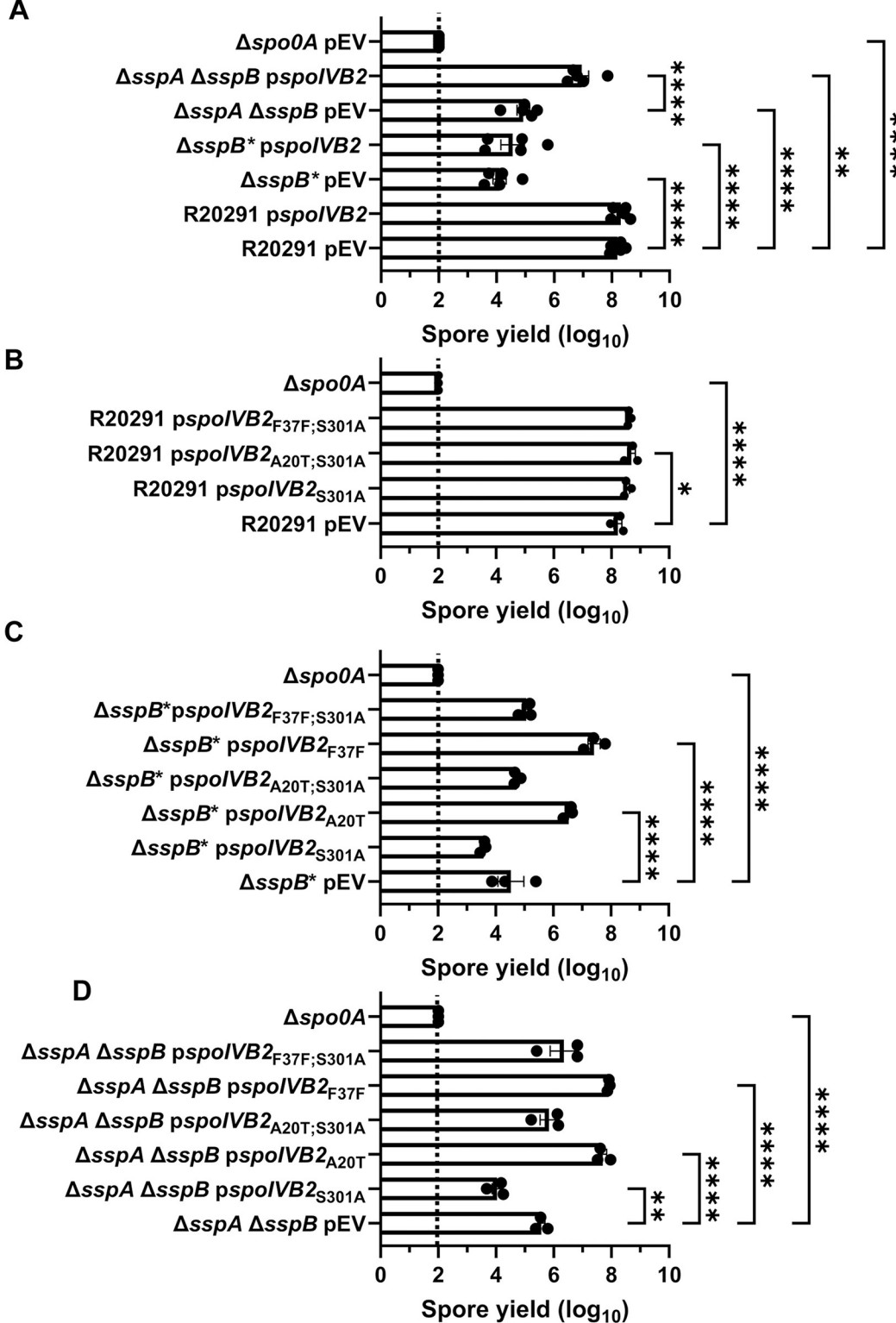

**Fig 4. Mutations in *spoIVB2* suppress the *C. difficile* *sspA* *sspB* mutant phenotype.** Sporulation was completed over 48 hours and the spore yield was determined. A) Sporulation of indicated strains with an empty vector (pEV) or wild type *spoIVB2* B) Sporulation of *C. difficile* R20291 with the indicated *spoIVB2* allele expressed from a plasmid. C) Sporulation assay of *C. difficile* R20291 Δ*sspB** with the indicated *spoIVB2* allele expressed from a plasmid. D) Sporulation assay of R20291 Δ*sspA* Δ*sspB* with the indicated *spoIVB2* allele expressed from a plasmid. All data points represent the average from

at least three independent experiments. Statistical analysis by two-way ANOVA with Šídák's multiple comparisons test* P<0.05, ** P<0.01, *** P<0.001, **** P<0.0001.

When the *spoIVB2* alleles were introduced into the *C. difficile* Δ*sspB*\* strain, the *spoIVB2*$_{S301A}$ allele did not restore sporulation, but the *spoIVB2*$_{A20T}$ and *spoIVB2*$_{F37F}$ alleles increased sporulation by approximately 2 and 3-log$_{10}$, respectively. When these alleles were combined with the *spoIVB2*$_{S301A}$ allele, sporulation was not restored (Fig 4C). These results were similar to when the *spoIVB2* alleles were expressed in the *C. difficile* Δ*sspA* Δ*sspB* strain [the expression of *spoIVB2*$_{A20T}$ and *spoIVB2*$_{F37F}$ resulted in an approximate 2-log$_{10}$ increase in spore yield] (Fig 4D). The catalytically dead allele in combination with the identified alleles from EMS was again unable to restore sporulation. These results suggest that the catalytic activity of *C. difficile* SpoIVB2 is important for its function.

From these data, we hypothesized that SspA and SspB directly or indirectly activate the expression of *spoIVB2* in wildtype cells in the late stages of sporulation. Thus, in the *C. difficile* Δ*sspA* Δ*sspB* and the *C. difficile* Δ*sspB*\* mutant backgrounds, SpoIVB2 levels are reduced, resulting in the failure to proteolytically activate, or inactivate, a target that is essential for sporulation. The expression of wild type *spoIVB2* (Fig 4A) does not greatly restore the sporulation deficient phenotype, likely because it is not expressed during σ$^G$ gene activation. The suppressor strains potentially increase the amount of SpoIVB2 present, bypassing the need for σ$^G$-dependent induction of the *spoIVB2* transcript. To further evaluate this, we expressed wild type *spoIVB2* (from a plasmid) in the suppressor strains that have *spoIVB2*$_{A20T}$ or *spoIVB2*$_{F37F}$ and quantified spore formation. We found no significant difference in spore yield between the suppressor strains with an empty vector and those expressing wild type *spoIVB2* from its native promoter (S2A Fig). Furthermore, we generated clean strains containing *spoIVB2*$_{A20T}$ or *spoIVB2*$_{F37F}$ in the wild type or the *C. difficile* Δ*sspA* Δ*sspB* strains to eliminate from analysis the outside mutations from EMS treatment. The *spoIVB2* alleles in the wild type background, with an empty vector or a vector expressing wild type *spoIVB2*, did not impact the spore yield. However, the strains containing the identified *spoIVB2* alleles, with an empty vector or a vector expressing wild type *spoIVB2*, in the *C. difficile* Δ*sspA* Δ*sspB* strain increase spore yield 3-log$_{10}$, compared to *C. difficile* Δ*sspA* Δ*sspB* alone. Again, the addition of wild type *spoIVB2* did not further rescue the spore yield, indicating that the suppression is due to the altered *spoIVB2* alleles (S2B Fig).

### The C. difficile spoIVB2 mutant is phenotypically similar to the C. difficile ΔsspA ΔsspB strain

To further evaluate the role of SpoIVB2 during sporulation, we generated a deletion of *spoIVB2* in the *C. difficile* R20291 strain. The *C. difficile* Δ*spoIVB2* strain generated phase gray spores, similar to our observations for the *C. difficile* Δ*sspA* Δ*sspB* strain (Fig 5A). This phenotype could be complemented by expression of *spoIVB2*$_{WT}$, *spoIVB2*$_{A20T}$, or *spoIVB2*$_{F37F}$ alleles from a plasmid. However, restoration did not occur when the catalytically dead *spoIVB2*$_{S301A}$ was expressed (Fig 5A). The spore yield of the *C. difficile* Δ*spoIVB2* strain was 6-log$_{10}$ lower than wild type. The *C. difficile* Δ*spoIVB2* mutant supplemented with a plasmid expressing *spoIVB2* wild type, A20T or F37F alleles restored the spore yield to wild type levels (Fig 5B). However, when the S301A allele was present or in combination with the A20T or F37F alleles, sporulation was not restored, again highlighting the importance of catalytic activity in the function of SpoIVB2 (Fig 5B).

Analysis of the *C. difficile* Δ*spoIVB2* strain by TEM revealed deviations from the sporulating cells (Fig 6). As seen in the field of view image, it was difficult to locate whole cells for imaging.

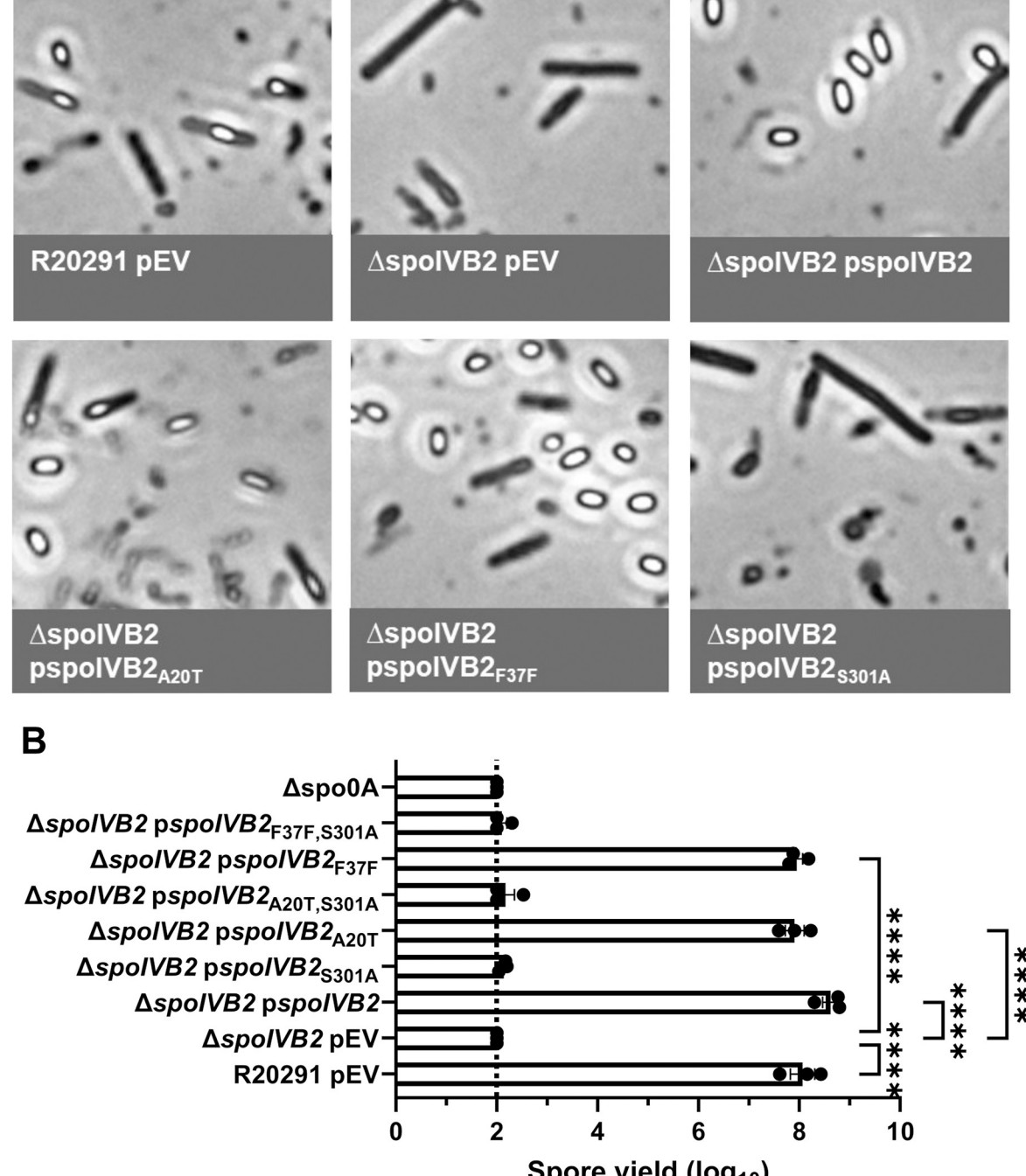

**Fig 5. *C. difficile* Δ*spoIVB2* has a sporulation defect.** A) Day 6 cultures were fixed in 4% formaldehyde and 2% glutaraldehyde in PBS and imaged on a Leica DM6B microscope. B) Spore yield of the indicated strain was determined as described in Fig 1. pEV indicates an empty plasmid within the strain. All data points represent the average from three independent experiments. Statistical analysis by one-way ANOVA with Šídák's multiple comparisons test. * $P<0.05$, ** $P<0.01$, *** $P<0.001$, **** $P<0.0001$.

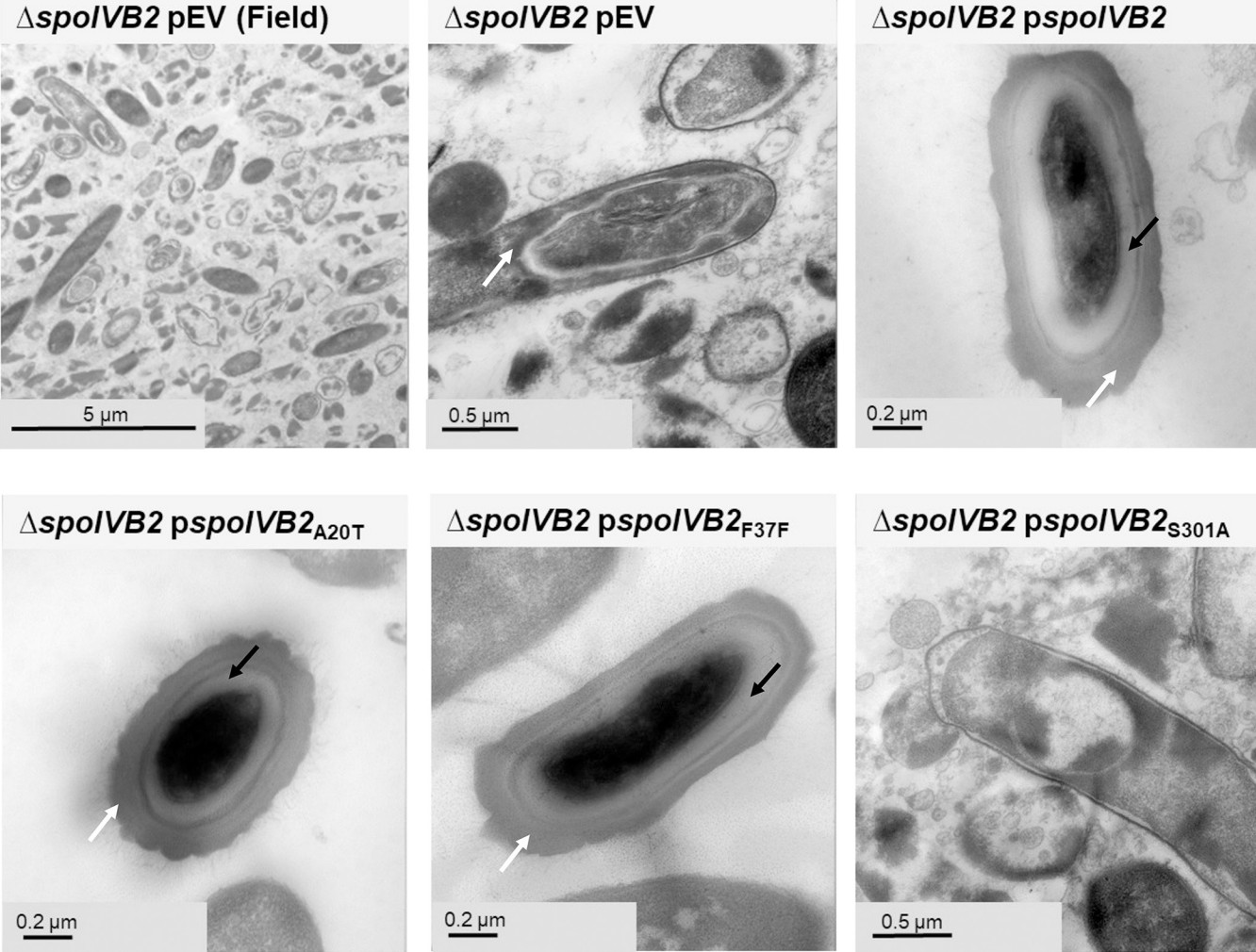

**Fig 6. SpoIVB2 is required for cortex synthesis.** Day 6 sporulating cultures were prepared for TEM. A field of view image is shown for the *C. difficile* Δ*spoIVB2* mutant while the remainder of the images are zoomed into the sporulating cell / spore. The coat layer is indicated with a white arrow, while the cortex layer is indicated with a black arrow. pEV indicates an empty plasmid within the strain.

When a sporulating cell was found, there were structural issues within the forespore. The cortex was missing and, with the lack of its constraint around the core, appeared to allow for the expansion of the core contents. Though the coat and exosporium were present, it appears anomalous with inconsistent coat deposition and the lack of the characteristic lamellar coat structures. When *spoIVB2*$_{WT}$, *spoIVB2*$_{A20T}$, or *spoIVB2*$_{F37F}$ were expressed from a plasmid, the structural appearance of the spore was restored to wild type. However, when *spoIVB2*$_{S301A}$ was expressed, it remained difficult to locate any sporulating cells.

## Testing the impact of the suppressor alleles on spoIVB2 expression

To understand how the SASPs influence *spoIVB2* and / or other gene transcripts, RNA was extracted from *C. difficile* wild type, *sspA* and *sspB* single and double mutants, the *sspB*\* and *spoIVB2* mutant strains, as well as two representative suppressor strains from EMS mutagenesis at 11 hours post plating on sporulation medium and RT-qPCR was performed. Overall, at this time point, there were few differences in transcript levels. The *spoIVB2*$_{A20T}$ isolate

(HNN19) was variable between extractions despite testing more biological replicates, potentially due to other mutations from the EMS treatment. Though *sspA* or *sspB* transcripts levels were largely unchanged, there was a slight increase in transcript levels in comparison to wild type for *spoIVA* which encodes a protein involved in spore coat localization (S3A–S3C Fig) [51]. Transcripts for *sleC*, *pdaA*, and *spoVT* remained similar to wild type levels (S3D–S3F Fig). *spoIVB* transcript levels did not have a concise trend while, for *spoIVB2*, the general trend was towards slightly reduced transcripts in the mutant strains with a larger fold change in the EMS identified alleles (S4A and S4B Fig). *spoIIP* transcripts were slightly elevated in the mutant strains, except for the EMS isolates (S4C Fig). For the DPA synthesis and packaging protein transcripts (*dpaA*, *spoVAC*, *spoVAD*, and *spoVAE*), there were minimal differences for the mutant strains besides a slight increase in *spoVAC* (S5A–S5D Fig).

## Manipulation of the F37 and F36 codons impact suppression

Next, we manipulated the F37 codon to see if other changes would allow for sporulation to be restored in the mutant strains. We also changed the F36 codon from UUU to UUC (generating an F36F silent mutation and the opposite codon change that occurred in the F37F allele). These constructs were expressed from a plasmid under the *spoIVB2* native promoter region, and the spore yield was assessed. When wild type *spoIVB2* (UUC codon) was expressed in the *C. difficile* Δ*sspB** strain, sporulation was not restored to wild type levels (Fig 7A). However, sporulation was partially restored with the *spoIVB2*$_{F37F}$ (UUU codon), the *spoIVB2*$_{F37L}$ (UUA codon), the *spoIVB2*$_{F37L}$ (UUG codon) and the *spoIVB2*$_{F36F}$ (UUC codon) alleles (Fig 7A). This suggests that multiple *spoIVB2* variants were sufficient to restore sporulation in an otherwise sporulation deficient strain.

Expression of these plasmids in the *C. difficile* Δ*sspA* Δ*sspB* double mutant strains showed variation from the previously assessed strain. First, expression of the wild type *spoIVB2* allele resulted in an approximate 1-$\log_{10}$ increase in spore yield compared to the mutant strain with an empty vector (Fig 7B). However, expression of the *spoIVB2*$_{F37F}$ (UUU codon) or the *spoIVB2*$_{F37L}$ (UUA or UUG codons) restored the spore yield to a higher level than the wild type *spoIVB2* allele. Interestingly, in the *C. difficile* Δ*sspA* Δ*sspB* strain, *spoIVB2*$_{F36F}$ did not complement the sporulation phenotype as it did in the *C. difficile* Δ*sspB** strain (Fig 7B). Finally, expression of any of the *spoIVB2* alleles restored sporulation in the *C. difficile* Δ*spoIVB2* mutant strain (Fig 7C). These data suggest that either altering the F37 codon in either of the sporulation deficient strains or expressing additional SpoIVB2 can restore sporulation.

## spoIVB2$_{A20T}$ and spoIVB2$_{F37F}$ have increased abundance

We next wanted to determine if the suppressor alleles restore sporulation through translational differences, rather than transcriptional, we designed a luciferase-based assay [10,52,53]. SpoIVB2 is a single span transmembrane protein whose C-terminus is located outside of the forespore cytoplasm. To the *spoIVB2* gene, we engineered a *ssrA* tag to the 3' end of the gene. This will tag the protein for degradation by the ClpP protease if the protein is in the cytoplasm but ClpP will not have access to the C-terminus if it is localized properly [54]. This assay will allow us to quantify differences in properly-localized SpoIVB2. As a control, the *bitLuc* gene with and without the *ssrA* tag was put under control of the native *spoIVB2* promoter. We also coupled the native *spoIVB2* promoter to the *bitLuc* gene and either wild type *spoIVB2*, *spoIVB2*$_{A20T}$, or *spoIVB2*$_{F37F}$ and tagged the construct for degradation with a *ssrA* tag. These constructs were introduced into the wild type and, as a negative control, the *C. difficile* Δ*spo0A* strain. When in *C. difficile* Δ*spo0A*, all constructs had minimal RLU/OD$_{600}$ values. After

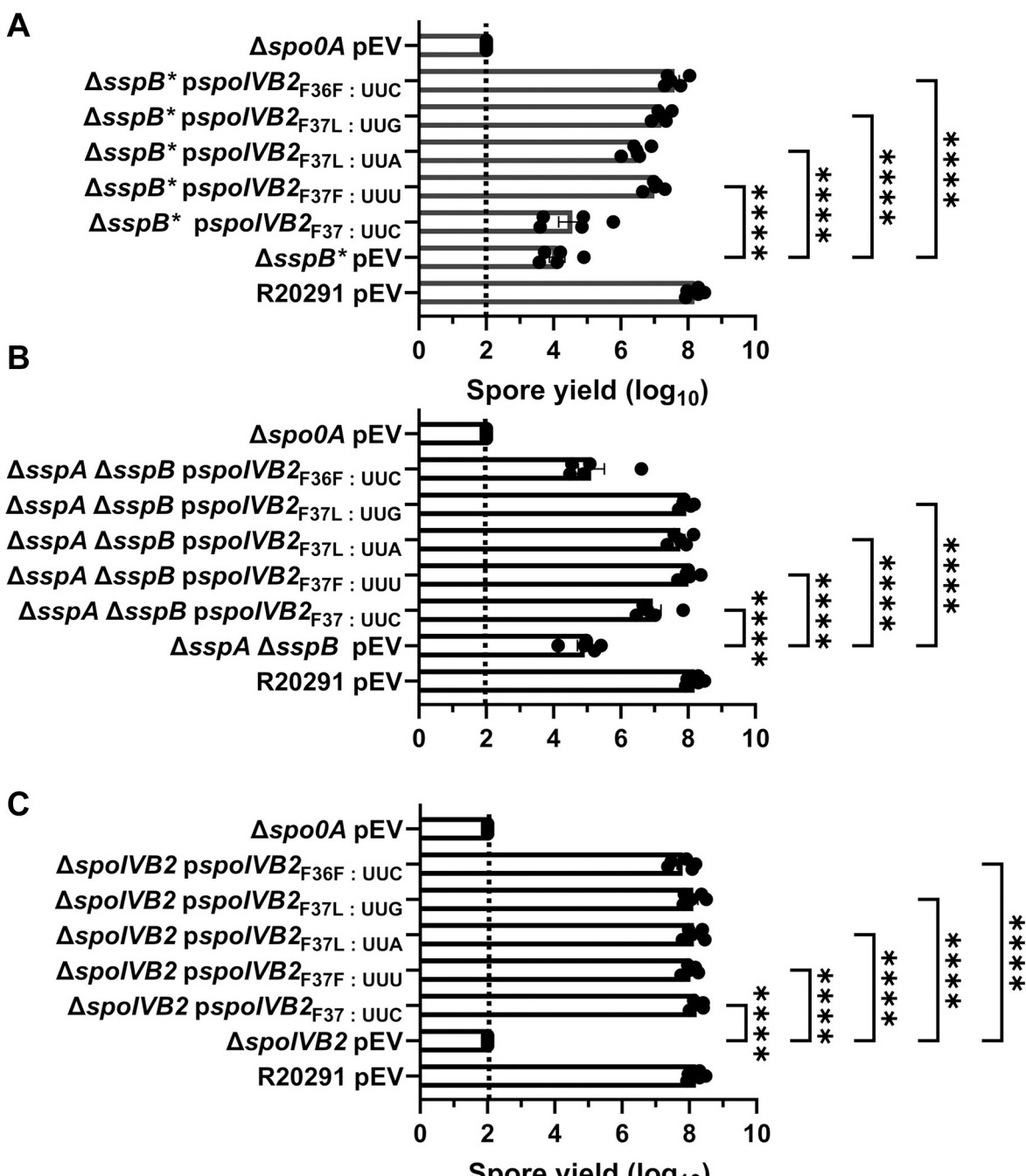

**Fig 7. Manipulation of *spoIVB2*<sub>F36 / F37</sub> restores sporulation.** Spore yield of the indicated strain was determined as described in Fig 1. The plasmids expressing the *spoIVB2*$_{F37}$ alleles and the *spoIVB2*$_{F36F}$ allele were assessed in the strains A) *C. difficile* Δ*sspB**, B) *C. difficile* Δ*sspA* Δ*sspB*, C) *C. difficile* Δ*spoIVB2*. pEV indicates an empty plasmid within the strain. All data represents the average of five independent experiments. Statistical analysis by ANOVA with Šídák's multiple comparison test. * P<0.05, ** P<0.01, *** P<0.001, **** P<0.0001.

expression in the wild type strain, the control construct containing the *ssrA* tag had significantly lower normalized luminescence / $OD_{600}$ than the construct without the tag (Fig 8). This shows that the *ssrA* tag successfully reduced luciferase abundance. After 48 hours of incubation in the wild type strain, the *spoIVB2*$_{A20T}$ construct had 800x greater levels of luminescence /

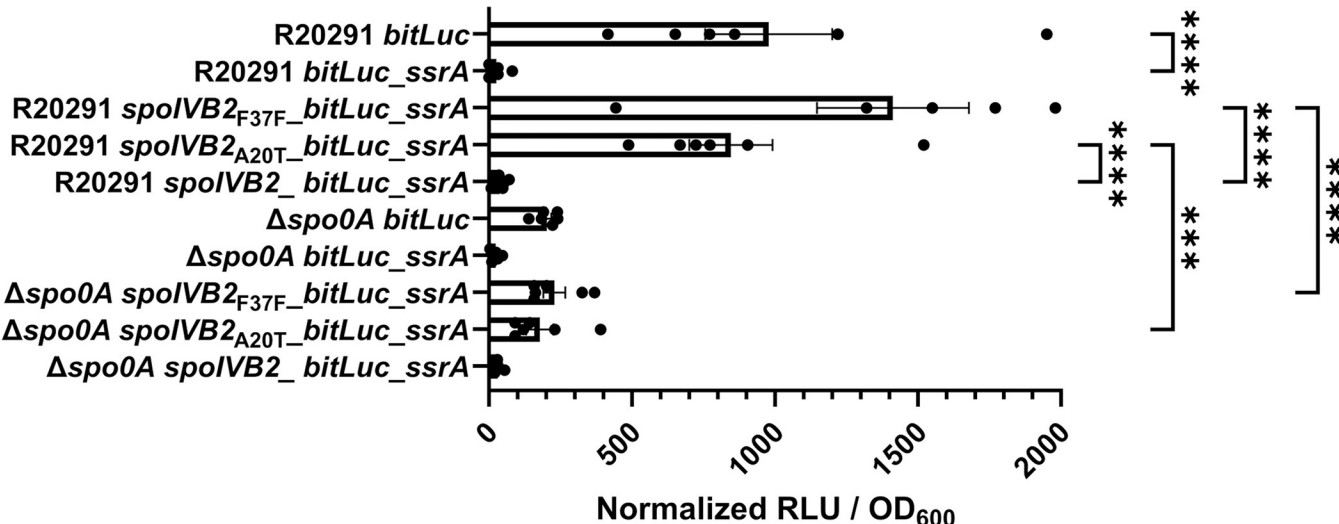

**Fig 8. Luciferase assays show minimal differences between *spoIVB2* protein levels.** Steady-state protein levels of the indicated strain were determined using alleles with an engineered *ssrA* tag (as described in the materials and methods). The cultures were grown for 48 hours and the OD$_{600}$ and the RLUs were determined. The RLUs were normalized to the OD$_{600}$ of each culture. All data represents the average of six independent experiments. One data point in the R20291 *spoIVB2*$_{F37F}$_*bitLuc_ssrA* data set was found to be an outlier by the ROUT test and was removed from analysis. Statistical analysis by ANOVA with Šídák's multiple comparison test. * P<0.05, ** P<0.01, *** P<0.001, **** P<0.0001.

OD$_{600}$ compared to wild type and the *spoIVB2*$_{F37F}$ construct had approximately 1,300x greater levels (Fig 8). These data suggest that sporulation is restored in the suppressor strains because the identified alleles increased the abundance of SpoIVB2 that was present in the sporulating cell.

## Restoration of sporulation using different promoters to drive spoIVB2 expression

To understand if the SASPs allow for continued *spoIVB2* expression during σ$^G$ gene activation, we generated plasmids containing *spoIVB2* expressed by various promoters. The *spoIVB* promoter region served as a lower activity σ$^G$ promoter while the *sspA* promoter region served as a higher activity σ$^G$ promoter. The spore yield of the *C. difficile* Δ*spoIVB2* strain was rescued when *spoIVB2* was expressed under the *spoIVB2*, *spoIVB*, or the combined *spoIVB* / *spoIVB2* promoters, suggesting that SpoIVB2 can be present during later stage sporulation (Fig 9B). However, *spoIVB2* expressed under the *sspA* or the combined *spoIVB2* / *sspA* promoters did not restore sporulation. Interestingly, the *spoIVB2* / *sspA* promoter combination when in wild type cells also reduced spore yield 5-log$_{10}$ (Fig 9A). Similarly, spore yield in *C. difficile* Δ*sspA* Δ*sspB* was restored when *spoIVB2* was expressed under the *spoIVB1*, *spoIVB2*, or the *spoIVB1* and *spoIVB2* combined promoters (Fig 9C). When expressed under the *sspA* or the combined *spoIVB2* and *sspA* promoters, restoration did not occur. We hypothesize that the highly active *sspA* promoter leads to overproduction of SpoIVB2, which is then detrimental to the sporulating cells.

## Discussion

The formation of endospores in *C. difficile* is vital for transmission of disease and the mechanisms involving spore formation are complex [8]. In prior work, we determined that the *C. difficile* Δ*sspA* Δ*sspB* strain was halted during sporulation suggesting that the *C. difficile* SASPs are important for regulating late-stage sporulation, somehow [42]. Here, we built upon our

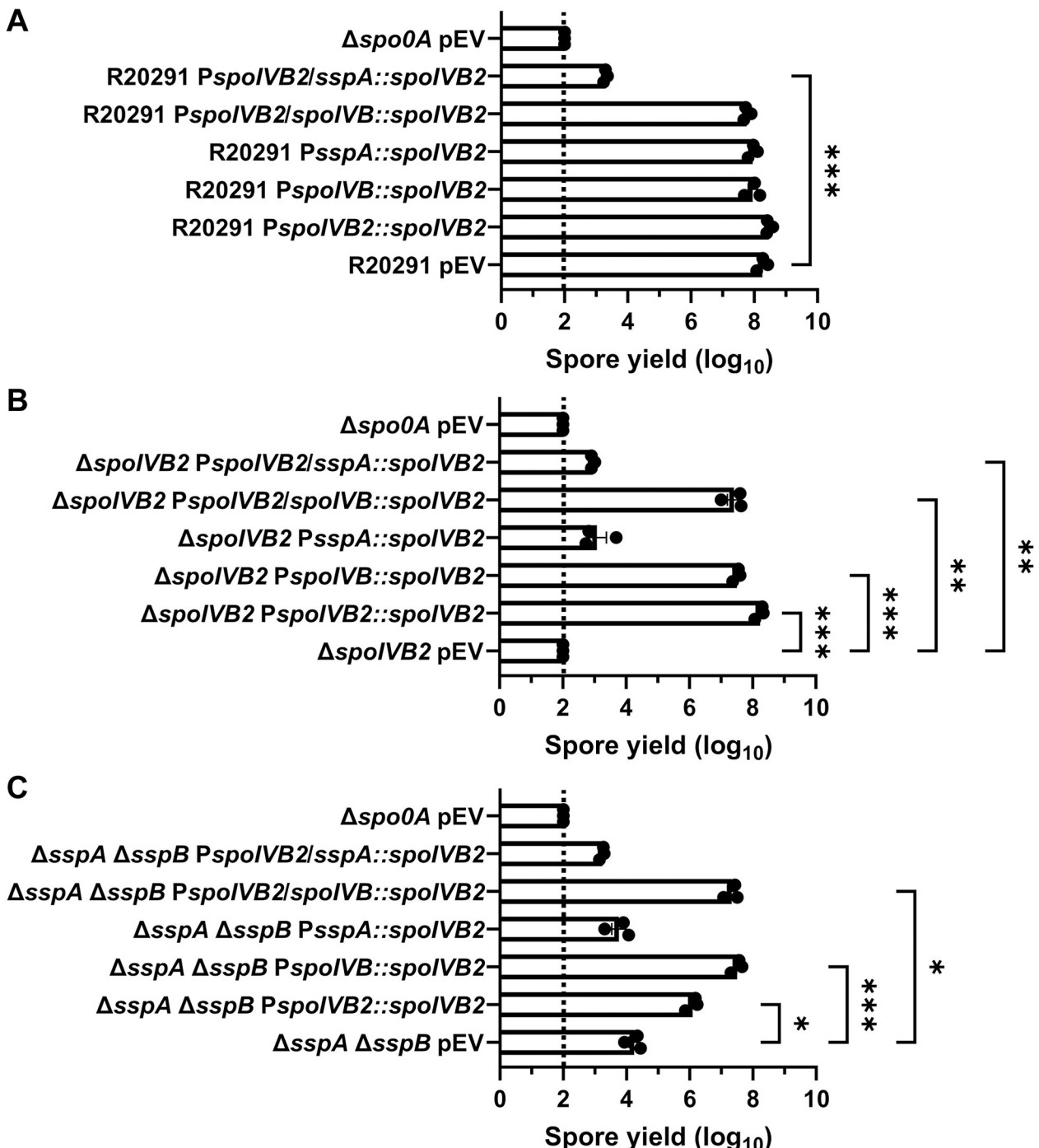

**Fig 9. Alternative promoters driving *spoIVB2* rescues the sporulation phenotype.** Plasmids expressing *spoIVB2* under various promoters were expressed in either wild type, the *C. difficile* Δ*spoIVB2*, or the *C. difficile* Δ*sspA* Δ*sspB* strains. Spore yield was determined after 48 hours of incubation by treatment with 30% EtOH and plating on medium containing germinants. The CFUs were $\log_{10}$ transformed. All data represents the average of three independent experiments. Statistical analysis by ANOVA with Šídák's multiple comparison test. * P<0.05, ** P<0.01, *** P<0.001, **** P<0.0001.

findings by further exploring the SASP mutant strain using TEM and a selection strategy to identify potential suppressor mutants.

Oddly, during the course of the prior work, we identified a mutation in the *C. difficile sspA* gene during the generation of the *sspB* mutant using CRISPR-Cas9 editing. This strain, *C. difficile* Δ*sspB*; *sspA*$_{G52V}$ (*C. difficile* Δ*sspB\**), had a phenotype similar to the *C. difficile* Δ*sspA* Δ*sspB* strain. This phenotype was likely due to the missense mutation within a conserved glycine residue. Prior work in *B. subtilis* found that SspC$^{G52A}$ poorly bound DNA [37,55]. Oddly, we have since observed a similar off-target effect in the *sspB* gene when targeting *sspA* using CRISPR--Cas9 mutagenesis. During the process of targeting *sspA* in a *C. difficile* Δ*gpr* strain, an *sspB*$_{E64stp}$ allele was also observed upon confirmation of the mutant's DNA sequence. The two genes are not located in close proximity nor do the constructs for deletion encode this sequence. We hypothesize that there may be some selective pressure to mutate *sspA* or *sspB* within a deletion strain.

With further evaluation of SASP mutant strains by TEM, we found that the *C. difficile* Δ*sspA* Δ*sspB* strain produces forespores that are blocked after the engulfment step, and do not contain cortex. Cortex is synthesized under SpoVD and potentially SpoVE, control and is modified by PdaA, GerS, and CwlD proteins [56–61]. These proteins modify the peptidoglycan to generate muramic-δ-lactam residues [59–62]. The cortex provides a physical constraint around the spore core, maintaining size and preventing water from hydrating the Ca-DPA-rich core [8,63]. In the absence of cortex, it is likely that some contents in the spore core leak out. This likely explains our previous findings that the few spores that could be purified from the *C. difficile* Δ*sspA* Δ*sspB* and the *C. difficile* Δ*sspB\** strains contained little CaDPA [42]. Because our RT-qPCR data showed that *dpaA* and *spoVAC/D/E* transcript levels are similar to wild type levels in these mutant strains, it is likely that DPA is being synthesized and transported into the spore but cannot be concentrated into the core without a mature cortex layer.

An EMS mutagenesis strategy to find suppressors of the defect in sporulation of the *C. difficile* Δ*sspA* Δ*sspB* strain identified mutations in *spoIVB2*. SpoIVB2 is homologous to *B. subtillis* SpoIVB. Though *B. subtilis* contains the same spore layers and sigma factors that regulate sporulation in *C. difficile*, the process is more complex. The *B. subtilis* SpoIVB protease is produced under σ$^G$ control. Located in the *B. subtilis* outer forespore membrane are the SpoIVFB, SpoIVFA, and BofA proteins [64]. BofA is an inhibitor of the SpoIVFB protease, and SpoIVFA keeps the proteins localized in the membrane. SpoIVB is secreted through the inner forespore membrane and processes SpoIVFA, thereby relieving BofA inhibition of SpoIVFB. Activated SpoIVFB cleaves the pro-peptide from σ$^K$, resulting in σ$^K$ activation [64].

In addition to its role in σ$^K$ activation, SpoIVB has other functions in *B. subtilis*, *e.g.* cleavage of SpoIIQ [65]. SpoIIQ is required for σ$^G$ synthesis and contributes to the formation of a feeding tube between the mother cell and the forespore compartments. SpoIVB cleaves SpoIIQ upon completion of engulfment, however, this cleavage is not necessary for spore formation or any later-stage gene expression [65,66]. A *spoIVB* null mutant blocks the formation of fully formed, heat resistant spores [67]. Spores derived from this strain form the forespore but lack the germ cell wall layer and do not generate mature spores. Interestingly, this phenotype was independent of SpoIVB's role in the activation of σ$^K$ [67]. An alternative role for SpoIVB may be in germ cell wall biosynthesis or as a DNA binding regulatory protein.

Even though the *C. difficile* sporulation program does not contain the cross-talk sigma factor activation or homologs to *bofA*, *spoIVFA*, or *spoIVFB*, it does contain the SpoIVB and SpoIVB2 paralogs [28]. SpoIVB2 is σ$^F$-regulated while SpoIVB is σ$^G$-regulated [47]. *C. difficile* SpoIVB and SpoIVB2 contain 31% identity to each other and have 36% and 37% identity to *B. subtilis* SpoIVB, respectively [47].

In our sporulation assays, *spoIVB2*$_{A20T}$ and *spoIVB2*$_{F37F}$ can rescue the mutant phenotype and form mature, dormant spores. We hypothesize that the A20T and F37F alleles suppress the phenotype through translational changes. Interestingly, in the identified *spoIVB2*$_{F37F}$ strain, the wildtype UUC codon is used in 5.9 / 1000 codons but the UUU codon in the suppressor strain is used 37.4 / 1000 codons [68]. Also, out of the 18 phenylalanine residues found in the SpoIVB2 protein, only F37 uses the UUC codon. Even though the codon changes to one that is used more frequently, this data is based on codon usage across the whole *C. difficile* genome and not just spore specific genes.

When the wobble position of *spoIVB2*$_{F37}$ was manipulated, sporulation was restored in both, the *C. difficile* Δ*sspA* Δ*sspB* mutant and the *C. difficile* Δ*sspB\** mutant strains (even though the F37F allele was only identified in the former strain). Also of note, it is likely that the mutation to the UUU codon was the only identified change after EMS treatment, instead of the UUA or UUG codons, due to the nature of EMS mutagenesis which results in transition mutations. However, it is likely that the specific manipulations do not matter as long as the codon increases translation efficiency compared to the UUU codon. We analyzed transcript variation among strains for various genes, including *spoIVB2*. In this data, the transcripts for *spoIVB2* in representative EMS strains, for both the *spoIVB2*$_{A20T}$ and *spoIVB2*$_{F37F}$ alleles, trended toward being downregulated, though this difference was only ~4 fold and did not meet statistical significance (S4 Fig). These results support the hypothesis that SASPs are necessary to further activate *spoIVB2* transcription. Unfortunately, *C. difficile* sporulation is asynchronous and samples from any time point contain cells in every stage of sporulation. This could explain why the fold changes are small and variable across all strains and all transcripts analyzed. Because of this noise, it is difficult to draw definitive conclusions from the RT-qPCR data.

In our working model for how the *C. difficile* SASPs influence *spoIVB2*, *w*e hypothesize that SASP binding could directly activate gene expression by enhancing interactions with RNA polymerase. During the early stages of sporulation, *spoIVB2* expression is dependent upon σ$^F$. In wild type *C. difficile*, the σ$^G$-produced SASPS could further activate *spoIVB2* expression to maintain SpoIVB2 abundance in the spore during later stages of sporulation (Fig 10). In the absence of *C. difficile* *sspA* and *sspB*, SpoIVB2 activity is reduced in the later stages of sporulation, the spore is unable to synthesize cortex, and sporulation is halted (Fig 10). Since the SASPs are not present in the suppressor strains to lead to activated *spoIVB2* transcription, we hypothesized that the *spoIVB2* alleles identified alter translation rates and, thus, increase SpoIVB2 levels in the later stages of sporulation (Fig 10). Supporting this, our BitLuc data showed a significant increase in RLU for translation of the A20T and F37F alleles compared to wild type.

Separate from how the *spoIVB2* alleles restore sporulation to the SASP mutant strains, what is the function of SpoIVB2 during sporulation? While it is possible that the *C. difficile* SpoIVB and SpoIVB2 proteins retain a function in SpoIIQ cleavage, in *C. difficile*, SpoIIQ does not appear to be cleaved during *C. difficile* sporulation [69]. However, unlike in *B. subtilis*, *C. difficile* SpoIIP has a cleaved form that is only present in cells that have completed engulfment [69]. SpoIIP is an amidase and endopeptidase that works in concert with SpoIID to restructure peptidoglycan during forespore engulfment. In a *C. difficile* Δ*spoIIP* strain, the leading edge of engulfment does not progress, so engulfment does not occur [69]. In recent collaborative work, we found that *C. difficile* SpoIVB2 does cleave SpoIIP *in vitro* and *in vivo* [50]. We thought it possible that SpoIIP needs to be cleaved post engulfment completion to allow the following stages of sporulation to continue. However, we found that strains containing *spoIIP* with an altered cleavage site (SpoIVB2 is unable to cleave this form *in vitro*) could still form mature spores [50]. While it is possible that this altered SpoIIP could still be processed by

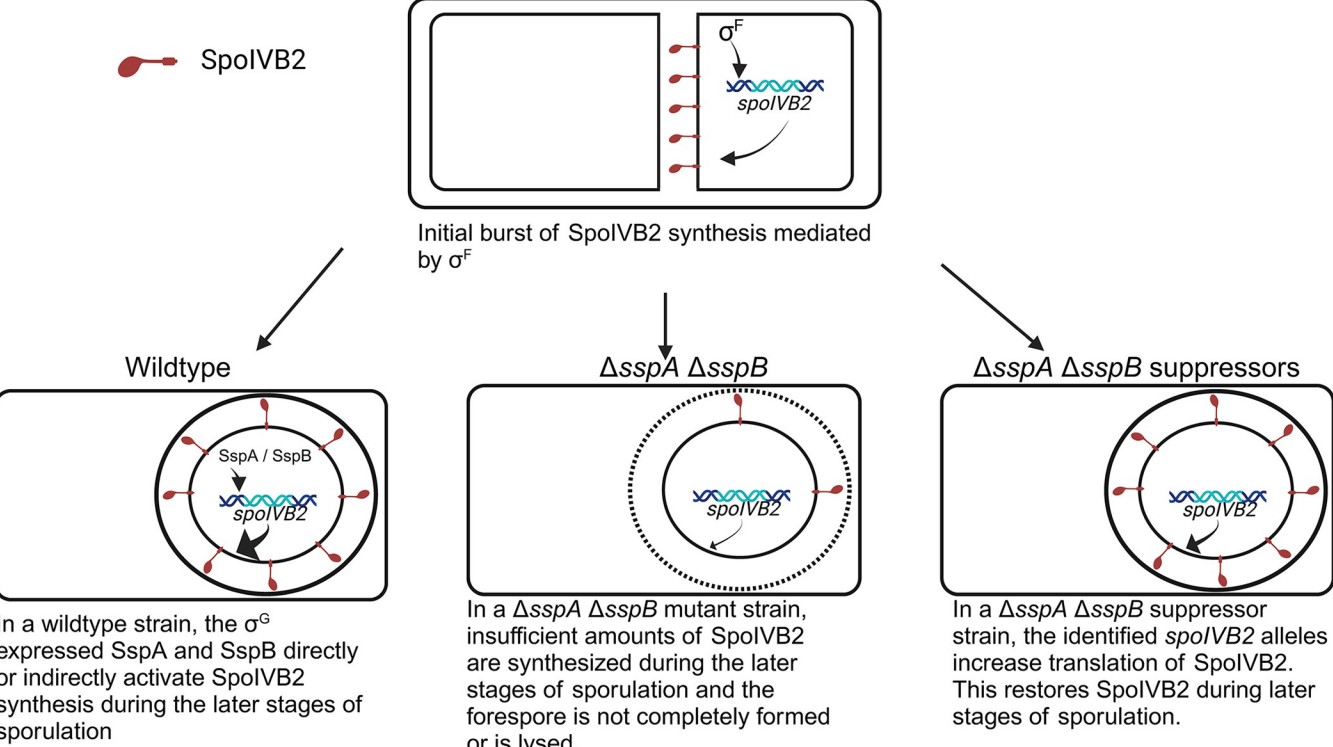

**Fig 10. Model for *spoIVB2* regulation by the *C. difficile* SASPs.** In a working model for how *C. difficile* SASPs regulate sporulation, SpoIVB2 is initially synthesized in sporulating cells under the control of σF. We hypothesize that upon σG activation in the forespore of wildtype *C. difficile* cells, SspA and SspB become expressed, and they directly or indirectly activate *spoIVB2* expression during the later stages of sporulation. In the *C. difficile* Δ*sspA* Δ*sspB* strain, insufficient SpoIVB2 is synthesized during the later stages of sporulation. This results in the inability to synthesize cortex and, thus, complete the sporulation program. In the *C. difficile* Δ*sspA* Δ*sspB* or in the *C. difficile* Δ*sspB** suppressor strains, the *spoIVB2*~*A20T*~ or the *spoIVB2*~*F37F*~ alleles lead to an increase in the translation of the *spoIVB2* transcript and results in a restoration of SpoIVB2 during the later stages of sporulation. Created with BioRender.com.

SpoIVB2, just highly inefficiently, it is likely that there are uncharacterized targets / functions of SpoIVB2 during sporulation.

To better understand if the SASPs function similarly among other organisms, we also tested the ability of the *B. subtilis sspA* gene to complement sporulation and UV phenotypes in *C. difficile* mutants. When expressed under the *C. difficile sspA* native promoter region, *B. subtilis sspA* can partially restore sporulation and UV resistance. In prior work from the Setlow lab [70], the authors suggested that SASPs may affect forespore transcription, likely by physically blocking RNA polymerase when binding in high concentrations. Furthermore, *in vitro* transcription assays in *B. subtilis* show that less *in vitro* transcription occurs when SASPs are incubated with DNA. However, transcription occurs in the absence of SASPs or in the presence of a SASP variant with poor DNA binding ability [37]. However, they also found evidence of some genes (mainly later stage sporulation genes) having higher / lower transcription in the *B. subtilis* Δ*sspA* Δ*sspB* strain [70]. These data indicate that the SASPs could be regulating sporulation in the forespore of both *C. difficile* and *B. subtilis*, suggesting that the different phenotypes observed between the Δ*sspA* Δ*sspB* double mutants in the two organisms lie in the differences between the mechanism of compartmental signaling during sporulation. This leads to further questions about whether the genes / regions of DNA that are influenced by the SASPs and how the SASPs may potentiate these affects differs between the two organisms.

Overall, this study gives insight into the sporulation process and regulation in *C. difficile*. It is likely that the SASPs have binding "hotspots" where in low concentrations they

preferentially bind to influence transcription. Although the RT-qPCR data did not show many transcriptional changes, we hypothesize that the SASPs are influencing transcription of target genes. It is possible that the time of extraction was not ideal for capturing transcriptional changes, the change is small enough that the variability in data due to sporulation being asynchronous could be enough to hide the effects, or, though unlikely based on our data, the SASPs have different targets than those tested. This study also highlights the importance of *C. difficile* SpoIVB2 during sporulation even though its exact role is still unknown. Further work needs to be completed to understand the influence of SpoIVB2 during sporulation and to determine other potential targets for the SASPs.

## Materials and methods

### Bacterial growth conditions

*C. difficile* strains were grown in a Coy anaerobic chamber at ~4% $H_2$, 5% $CO_2$, and balanced $N_2$ at 37˚C [71]. Strains were grown in / on brain heart infusion (BHI), BHI supplemented with 5 g / L yeast extract (BHIS), 70:30 (70% BHIS, 30% SMC) or tryptone yeast (TY) medium. 0.1% L-cysteine was added to BHI and BHIS while 0.1% thioglycolate was added to TY. Media was supplemented with thiamphenicol (10 μg / mL), taurocholate [TA] (0.1%), cycloserine (250 μg / mL), kanamycin (50 μg / mL), lincomycin (20 μg / mL), rifampicin (20 μg / mL), ethylmethane sulfonate [EMS] (1%), or D-xylose (0.5% or 1%) where indicated. *E. coli* strains were grown on LB at 37˚C and supplemented with chloramphenicol (20 μg / mL) or ampicillin (100 μg / mL). *B. subtilis* BS49 was grown on LB agar plates or in BHIS broth at 37˚C and supplemented with 2.5 μg / mL chloramphenicol or 5 μg / mL tetracycline.

### *Plasmid construction*: All cloning was performed in *E. coli* DH5α

For construction of the *C. difficile* CD630Δ*erm sspA*-targeting CRISPR vector, pHN120, 500 bp of upstream homology was amplified from CD630Δ*erm* genomic DNA with primers 5'sspA_MTL and 3'sspA_UP while the downstream homology arms were amplified with 5'sspA_down and 3' sspA_xylR. These were inserted into pKM197 at the *Not*I and *Xho*I sites using Gibson assembly [72]. The gRNA gBlock (Integrated DNA Technologies, Coralville, IA) CRISPR_sspA_165 was inserted at the *Kpn*I and *Mlu*I sites. pHN120 was then used as the base plasmid to change the *Tn916* oriT for the *traJ* oriT at the *Apa*I sites, resulting in pHN131. *traJ* was amplified from pMTL84151 with primers 5'traJ and 3'traJ. The gRNA was then replaced with gBlock CRISPR_sspA_135 at the *Kpn*I and *Mlu*I restriction sites, generating pHN138, which was used to make the deletion.

   For generating the *C. difficile* CD630Δ*erm sspB* targeting CRISPR vector, pHN121, the upstream homology was amplified from CD630Δ*erm* genomic DNA with 5' sspB UP and 3' sspB UP and the downstream homology with 5' sspB DN and 3' sspB_xylR. These homology arms were inserted into pKM197 at *Not*I and *Xho*I restriction sites using Gibson assembly [72]. The gRNA gBlock CRISPR_sspB_144 was also inserted into the *Kpn*I and *Mlu*I sites. The oriT was changed from *Tn916* to *traJ* by amplifying *traJ* from pMTL84151 with 5'traJ and 3'traJ and inserting in the *apa*I sites to generate pHN132.

   For generating the *C. difficile* R20291 *spoIVB2* targeted CRISPR plasmid, the upstream homology arm was amplified from R20291 with primers 5' CDR20291_0714 UP and 3' CDR20291_0714 UP while the downstream was amplified with 5' CDR20291_0714 DN and 3' CDR20291_0714 DN. These were inserted into pKM197 at the *Not*I and *Xho*I sites using Gibson assembly [72]. The gRNA was amplified from pKM197 using primers CDR20291_0714 gRNA 3 and 3' gRNA_change. This fragment was inserted into the *Kpn*I and *Mlu*I sites using Gibson assembly, generating pHN157 [72].

Plasmid pHN149 was generated by amplifying the *traJ* oriT from pMTLYN4 with primers 5' tn916.traJ and 3'traJ and the *Tn916* oriT from pJS116 with 5'Tn916ori_gibson and 3' tn916.traJ. These were inserted into the A*pa*I site of pMTL84151.

For the generation of pHN122, pHN123, and pHN127, the *spoIVB2* gene and promoter regions were amplified from HNN37, HNN19, and R20291, respectively, using 5' CDR20291_0714 and 3' CDR20291_0714. These fragments were inserted into pJS116 (for pHN122 and pHN123) or pHN149 (for pHN127) at the *Not*I and *Xho*I sites using Gibson assembly [72].

For pHN145, pHN146, and pHN147, the first segment of DNA was amplified from R20291, pHN122, or pHN123, respectively, using 5' CDR20291_0714 and 3' 0714_S301A. The second segment of DNA was amplified from pHN127 for all 3 plasmids using 5' 0714_S301A and 3' CDR20291_0714. These two fragments were inserted using Gibson assembly into pJS116 at the *Not*I and *Xho*I sites [72].

The CD630Δ*erm sspA* gene and promoter region were amplified from CD630Δ*erm* with the primers 5'sspA_MTL and 3' sspA.pJS116. This fragment was inserted into pMTL84151 at the *Not*I and *Xho*I sites using Gibson assembly, generating pHN152 [72].

For pHN153, the CD630Δ*erm sspA* gene and promoter region were amplified with 5'sspA_MTL and 3' sspAsspB. The CD630Δ*erm sspB* gene and promoter region were amplified with 5' sspAsspB and 3'sspBpJS116. These fragments were inserted into pMTL84151 at the *Not*I and *Xho*I sites using Gibson assembly [72].

The CD630Δ*erm sspB* gene and promoter region were amplified using 5' sspB UP and 3'sspBpJS116. This fragment was inserted into pHN149 at the *Not*I and *Xho*I sites using Gibson assembly to generate pHN176 [72].

pHN271 and pHN272, the *spoIVB2*$_{A20T}$ / *spoIVB2*$_{F37F}$ theophylline allelic exchange plasmids, respectively, were generated by amplifying the *spoIVB2* region with homology from pHN123 for pHN271 and pHN122 from pHN272 with primers 5' spoIVB2_theo and 3' spoIVB2_theo. These fragments were inserted into pJB94 at the *Not*I and *Xho*I sites using Gibson assembly [72, 73].

pJB96 was generated by amplifying the *sacB* gene from pJB94 with primers 5'sacB_UP and sacB_3'_XhoI and inserted into pHN149 at the *Not*I and *Xho*I restriction sites by Gibson assembly [72].

The *spoIVB2* under *sspA* promoter control plasmid, pHN312, was generated by amplifying the *sspA* promoter from R20291 with the primers 5'sspA_MTL and 3' PsspA_spoIVB2. The *spoIVB2* gene was amplified from R20291 with the primers 5' spoIVB2_PsspA and 3' CR20291_0714. These fragments were cloned into pJB96 at the *Not*I and *Xho*I sites by Gibson assembly. pHN329, *spoIVB2* under the *spoIVB* promoter, was made by amplifying the *spoIVB* promoter from R20291 with primers 5' spoIVB.pHN149 and 3' PspoIVB_spoIVB2. The spoIVB2 gene was amplified with 5' spoIVB2_PspoIVB and 3' CDR20291_0714 from R20291 template DNA. The fragments were assembled with Gibson assembly into pJB96 at *Not*I and *Xho*I cut sites [72]. The plasmid pHN330 containing *spoIVB2* driven by the *spoIVB* and *spoIVB2* promoters was generated by amplifying the *spoIVB2* promoter from R20921 DNA with primers 5' PspoIVB2_pHN149 and 3' PspoIVB(100)_PspoIVB2. The *spoIVB* promoter was amplified with primers 5' PspoIVB2_PspoIVB and 3' PspoIVB_spoIVB2 from R20291 DNA. The *spoIVB2* gene was amplified from R20291 with primers 5' spoIVB2_PspoIVB and 3' CDR20291_0714. These fragments were assembled into pJB96 at *Not*I and *Xho*I cut sites using Gibson assembly [72]. The plasmid pHN331 containing *spoIVB2* driven by the *sspA* and *spoIVB2* promoters was generated by amplifying the *spoIVB2* promoter from R20921 DNA with primers 5' PspoIVB2_pHN149 and 3' PsspA_PspoIVB2. The *sspA* promoter was amplified with primers 5' PsspA_PspoIVB2 and 3' PsspA_spoIVB2 from R20291 DNA. The *spoIVB2*

gene was amplified from R20291 with primers 5' spoIVB2_PsspA and 3' CDR20291_0714. These fragments were assembled into pJB96 at *Not*I and *Xho*I cut sites using Gibson assembly [72].

The luciferase plasmids pHN335 through pHN337 were generated by amplifying the *spoIVB2* promoter region from R20291 using primers 5' CDR20291_0714 and 3' spoIVB2_homol. The *spoIVB2* gene fragments were amplified with primers 5' spoIVB2_gene_homol and 3' spoIVB2end_lrgBit from R20291 for pHN335, pHN123 for pHN336, and pHN122 for pHN337. The *bitLuc* gene fragment with a *ssrA* tag was amplified from pMB81 with primers 5' lrgBit_spoIVB2end and 3' luciferase_ssrA_pHN149. These 3 fragments were cloned into pJB96 at the *Not*I and *Xho*I restriction sites using Gibson assembly [72]. For the control luciferase plasmids, pHN338-339, the *spoIVB2* promoter region was amplified from R20291 using primers 5 CDR20291_0714 and 3' spoIVB2_bitLuc. For pHN338, the *bitLuc* gene portion with a *ssrA* tag was amplified from pMB81 with primers 5' bitLuc_PspoIVB2 and 3' luciferase_ssrA_pHN149. For pHN339, the *bitLuc* gene was amplified from pMB81 with primers 5' bitLuc_PspoIVB2 and 3' luciferase_pHN149. The *spoIVB2* promoter fragment and the luciferase fragments were cloned into pJB96 at the *Not*I and *Xho*I restriction sites using Gibson assembly [72].

pHN220 was generated by amplifying the *sspA* promoter region from *C. difficile* R20291 with primers 5'sspA_MTL and 3'PsspA_BS49. The *sspA* gene was amplified from *B. subtillis* BS49 with primers 5' sspA_BS49 and 3' sspA_BS49. These fragments were put into the pHN149 backbone at the *Not*I and *Xho*I sites by Gibson assembly [72].

To generate pHN208, the promoter region through F36 of *CDR20291_0714* was amplified with 5' CDR20291_0714 and 3' spoIVB2 F36F, while the F36 through the end of *CDR20291_0714* was amplified with 5' spoIVB2 F36F and 3' CDR20291_0714, both using pHN127 as the DNA template. These fragments were inserted by Gibson assembly into the pHN149 plasmid backbone at the *Not*I and *Xho*I sites [72]. pHN218 was generated by amplifying from the pHN127 template DNA *CDR20291_0714* promoter region through F37 with primers 5' CDR20291_0714 and 3' spoIVB2 F37.UUA and the fragment with F37 through the end of the gene was amplified by 5' spoIVB2 F37.UUA and 3' CDR20291_0714. These fragments were inserted in pHN149 at the *Not*I and *Xho*I sites through Gibson assembly [72]. Similarly, pHN219 was generated but used 5' CDR20291_0714 with 3' spoIVB2 F37.UUG for the first fragment and 5' spoIVB2 F37.UUG and 3' CDR0291_0714 for the second fragment, both with pHN127 as the template DNA. These were also inserted by Gibson assembly into pHN149 at the *Not*I and *Xho*I sites [72].

All plasmids were sequenced to confirm the correct sequence of the inserts.

**Conjugations.** For conjugations between *C. difficile* and *E. coli* HB101 pRK24, 5 mL of LB supplemented with chloramphenicol and ampicillin was inoculated with a colony from the HB101 pRK24 transformation. Concurrently, *C. difficile* strains were cultured in 5 mL BHIS broth. After approximately 16 hours of incubation, 1 mL of *C. difficile* culture was heat shocked at 52˚C for 5 minutes, in the anaerobic chamber, and then allowed to cool. While heat shocking, 1 mL of *E. coli* culture was centrifuged at 2,348 x g for 5 minutes and the supernatant poured off. The *E. coli* pellets were passed into the chamber and resuspended with the cooled *C. difficile* culture. 20 μL spots were plated onto BHI. The next day, growth was scraped into 1 mL BHIS broth and distributed onto BHIS plates supplemented with thiamphenicol, kanamycin, and cycloserine (TKC) or TKC plus lincomycin (TKLC) for the 2-plasmid CRISPR system.

For conjugations between *C. difficile* and *B. subtilis* BS49, the plasmids generated in DH5α were used to transform *E. coli* MB3436 (a *recA*+ strain of *E. coli*) and plasmid purified. This plasmid preparation was then used to transform BS49. *C. difficile* was cultured in 5 mL BHIS

broth overnight. After approximately 16 hours, the *C. difficile* culture was back diluted 1:20 and grown for 4 hours. *B. subtilis* was grown for 4 hours in 5 mL BHIS broth supplemented with chloramphenicol and tetracycline. After incubation, the *B. subtilis* cultures were passed into the chamber and 100 μL of BS49 and 100 μL of *C. difficile* were spread onto TY plates. The next day, the growth was resuspended in 1 mL of BHIS broth and distributed between BHIS plates with thiamphenicol and kanamycin. Colonies were screened by streaking colonies onto BHIS supplemented with thiamphenicol and kanamycin and to BHIS supplemented with tetracycline to identify isolates that were tet-sensitive (do not contain the *Tn*916 transposon).

PCR was used to confirm strains and plasmids in each conjugate.

**CRISPR induction.**   For induction, colonies were passaged on TY agar supplemented with 1% xylose and thiamphenicol [42,74]. Mutants were detected by PCR and the plasmid was cured by heat shocking overnight cultures and isolating colonies that had lost their antibiotic resistance.

Induction of R20291 pHN138 resulted in the *C. difficile* CD630Δ*erm* Δ*sspA* mutant, HNN45. R20291 pHN132 induction resulted in the *C. difficile* CD630Δ*erm* Δ*sspB* mutant, HNN43. To generate the *C. difficile* CD630Δ*erm* Δ*sspA* Δ*sspB* strain, the pHN132 vector was induced in HNN45, resulting in HNN46. The *C. difficile* CDR20291_0714 (*spoIVB2*) mutant HNN49 was produced from induction of R20291 pHN157.

**Theophylline allelic exchange.**   Strains were generated as previously described [73]. Briefly, transconjugants were passaged on medium with thiamphenicol and no theophylline to encourage integration of the plasmid into the chromosome. Once integration occurred, the isolates were passaged on plates containing theophylline to encourage excision. HNN57 was generated from the passaging of R20291 pHN272. HNN60 was generated from the passaging of R20291 pHN271. HNN64 was generated from the passaging of *C. difficile* Δ*sspA* Δ*sspB* pHN271. HNN57 was generated from the passaging of *C. difficile* Δ*sspA* Δ*sspB* pHN272.

**Phase contrast imaging.**   The strains were inoculated onto 70:30 sporulation media and incubated for 6 days. After, the samples were fixed in a 4% paraformaldehyde and 2% glutaraldehyde solution in 1x PBS. The samples were imaged on a Leica DM6B microscope at the Texas A&M University Microscopy and Imaging Center Core Facility (RRID:SCR_022128).

**TEM.**   For sporulating cells, the relevant strains were incubated in the anaerobic chamber on sporulation media for 6 days, and then the growth scraped with an inoculation loop into 1,950 μL of fixative (5% glutaraldehyde, 2% acrolein in 0.05 M HEPES, pH 7.4) in a 2.0 mL microcentrifuge tube. The samples were incubated in the fixative overnight at 4°C. The following day, the fixed samples were centrifuged for 5 min at $>14,000 \times g$, and post-fixed and stained for 2 hours with 1% osmium tetroxide in 0.05 M HEPES, pH 7.4.

The samples were then centrifuged and washed with water 5 times, and dehydrated with acetone, using the following protocol: 15 minutes in 30%, 50%, 70%, 90% acetone each, then 100% acetone 3 changes, 30 minutes each. At the final wash step, a small amount of acetone, barely covering the pellets, was retained to avoid rehydration of the samples. The samples were then infiltrated with modified Spurr's resin (Quetol ERL 4221 resin, Electron Microscopy Sciences, RT 14300) in a Pelco Biowave processor (Ted Pella, Inc.), as follows: 1:1 acetone:resin for 10 minutes at 200 W–no vacuum, 1:1 acetone:resin for 5 minutes at 200 W–vacuum 20" Hg (vacuum cycles with open sample container caps), 1:2 acetone:resin for 5 minutes at 200 W–vacuum 20" Hg, 4 x 100% resin for 5 minutes at 200 W–vacuum 20" Hg.

The resin was then removed, and the sample fragments transferred to BEEM conical tip capsules prefilled with a small amount of fresh resin, resin added to fill the capsule, and capsules left to stand upright for 30 minutes to ensure that the samples sank to the bottom. The samples were then polymerized at 65°C for 48 hours in the oven, then left at RT for 24 hours before sectioning. 70–80 nm samples were sectioned by Leica UC / FC7 ultra-microtome

(Leica Microsystems), deposited onto 300 mesh copper grids, stained with uranyl acetate / lead citrate and imaged. All ultrathin TEM sections were imaged on JEOL 1200 EX TEM (JEOL, Ltd.) at 100 kV, images were recorded on SIA-15C CCD (Scientific Instruments and Applications) camera at the resolution of 2721 x 3233 pixels using MaxImDL software (Diffraction Limited). Images were subsequently adjusted for brightness / contrast using Fiji [75]. All equipment used is located at Texas A&M University Microscopy and Imaging Center Core Facility (RRID:SCR_022128).

**Sporulation assay.** Sporulation assays were completed as previously described [42]. Briefly, 70:30 plates were inoculated with the indicated strains and grown for 48 hours. 1/3 of the plate was harvested into 1 mL of PBS. 500 μL of the culture was treated for 20 minutes with 300 μL of 100% EtOH and 200 μL of $dH_2O$ to make a 30% final solution. After incubation, the samples were serially diluted in PBS + 0.1% TA and plated onto BHIS supplemented with TA to enumerate spores. The CFUs derived from spores were $log_{10}$ transformed.

**Spore purification.** Spores were purified as previously described [46, 76]. Briefly, the cultures from 70:30 agar medium were scraped into 1 mL of $dH_2O$ and left overnight at 4˚C. The next day, the pellets were resuspended and centrifuged for 1 minute at max speed. The upper layer of cell debris was removed and the sample was resuspended in 1 mL $dH_2O$. Again, the tubes were centrifuged and the upper layer removed. This was repeated approximately 5 times until the spore pellet was relatively free of debris. The 1 mL of spores in $dH_2O$ was loaded onto 50% sucrose and centrifuged at 4,000 x g for 20 minutes 4˚C. The spore pellet was then washed as described above approximately 5 times and then stored at 4˚C until future use.

**UV exposure.** UV experiments were performed as previously described [42]. Briefly, $1x10^8$ spores / mL in PBS were treated for 10 minutes with constant agitation. The $T_0$ and $T_{10}$ samples were serially diluted and plated onto rich medium containing germinant taurocholic acid (TA). Treated spore counts were normalized to untreated and then this ratio was normalized to the ratio for wild type spores.

**EMS treatment.** For EMS treatment, the HNN04 or HNN05 strains with pJS116 were used to help prevent contamination by providing antibiotic selection. Overnight cultures were back diluted to $OD_{600}$ 0.05 in 15 mL of BHIS + Tm [44,45]. The cultures were grown to an $OD_{600}$ of 0.5. The culture was split into 2 tubes of 5 mL, each. One tube served as the negative control and one tube was treated with 1% EMS. The cultures were grown for 3 hours with vigorous shaking every 30 minutes (to keep the EMS in solution). The cultures were passed out of the chamber and centrifuged at 3,000 x g for 10 minutes, passed into the chamber, decanted, resuspended with 10 mL BHIS to wash and then passed out and centrifuged again. This wash step was repeated 1 more time for a total of 2 washes. After the second wash, the cell pellet was resuspended with 1 mL of BHIS and deposited into 39 mL BHIS + Tm to recover overnight.

The next day, to determine mutagenesis rates, 10 μL, 25 μL, and 50 μL volumes were each plated onto BHIS rifampicin agar and CFUs were counted after 24–48 hours. From the EMS (+) culture, 50 μL was plated onto 20 BHIS Tm5 agar plates and left in the chamber to incubate for 5 days. For the EMS (-) culture, a whole genome prep was performed as described below.

After the incubation period, the plates were scraped into individual tubes with 1 mL $dH_2O$ and left overnight at 4˚C. The tubes were purified to remove cell debris as done with spore purification described above. The samples were combined to one tube and heated at 65˚C for 1 hour, with intermittent vortexing. The sample was then distributed between 20 BHIS Tm5 plates for another round of incubation. This enrichment step was completed 3 times before isolates were selected and PCR was used to confirm the genotype (to confirm that wildtype contamination did not occur during the selection). After confirmation, the samples were plated onto 70:30 Tm5 and incubated for 5 days. These samples were then checked under a phase contrast microscope for spores. Genomic DNA was purified from samples that had

spores and sent for whole genome resequencing at Microbial Genome Sequencing Center (MiGS; Pittsburgh, PA).

**Whole genome preparation.** 4 tubes of 10 mL each were inoculated overnight for approximately 18 hours (or the 40 mL of culture from EMS (-) strains were used). The next day, the samples were centrifuged at 4,000 x g for 10 minutes, 4°C. They were decanted, then resuspended with 1 mL TE buffer (10 mM Tris-HCl, 1 mM EDTA). Samples were centrifuged again, decanted, and resuspended with 200 µL of genomic DNA solution (34% sucrose in TE buffer) and transferred to a 2 mL Eppendorf tube (for each strain, the 4 tubes are kept separate). The tubes were incubated at 37°C for 2 hours. Then, 100 µL of 20% Sarkosyl and 15 µL of 10 mg / mL RNase A were added to the sample and incubated at 37°C for 30 minutes. After this incubation, 15 µL of proteinase K solution was added and incubated 37°C for 30 minutes at. The samples were brought up to 600 µL with TE buffer.

600 µL of 25:24:1 phenol/chloroform/isoamyl alcohol was added to the samples and were rocked gently for 20 minutes. After the incubation, the samples were centrifuged for 10 minutes at max speed. The upper layer was transferred to a new tube with a cut pipette tip (so as not to shear the DNA) and 600 µL chloroform was added to the sample and rocked for another 20 minutes. The centrifugation, sample transfer, and chloroform treatment were repeated for a total of 3 times. After which, the upper phase was transferred to a new tube and precipitated at -20°C overnight with 50 µL of 3 M sodium acetate, pH 5.2, and 3 volumes of cold 95% ethanol.

After precipitation, one tube from each strain was centrifuged 15 minutes at max speed, 4°C. The supernatant was discarded and the solution from the second tube was transferred to the tube with the DNA pellet and centrifuged again. This was repeated until the DNA from all 4 tubes was combined into one pellet. The DNA pellet was washed with 500 µL of 70% ethanol and centrifuged again. The samples were decanted and allowed to dry at room temperature until all of the ethanol was evaporated (approximately 60–90 minutes). After drying, 500 µL of either dH$_2$O or TE buffer was added, and the samples were rocked overnight to allow the pellets to dissolve.

**RNA extraction and processing.** Strains were plated onto 70:30 media for 11 hours before extraction. RNA extraction was performed using the FastRNA Pro Blue Kit (MP Biomedicals, Solon, OH). Briefly, the culture was scraped into 1 mL of PBS and centrifuged 2,348 x g for 5 minutes. The pellet was resuspended in 1 mL of RNApro solution and transferred to the provided tubes with lysing Matrix B. The cells were lysed in an MP FastPrep-24 bead beater for 40 seconds on and 20 seconds off for a total of 2 rounds. Further processing followed the FastRNA Pro Blue Kit protocol except that the RNA was precipitated overnight, and the remainder of the protocol was continued the next day.

Contaminating DNA was removed using the TURBO DNA-free kit (Invitrogen, Waltham, MA). 10 µg of RNA was treated 3 times with DNase following the protocol provided in the kit. The RNA was precipitated at -20°C overnight with 0.1 volume of 3 M sodium acetate, 5 µg of glycogen, and an equal volume of 100% ethanol. RNA was recovered by centrifuging at 13,000 x g at 4°C for 30 minutes. The pellet was washed 2 times with 70% cold ethanol. The pellet was air-dried at room temperature and then resuspended in dH$_2$O.

cDNA was generated using the Superscript III First-Strand Synthesis System (Invitrogen, Waltham, MA) reagents and protocol.

**RT-qPCR.** qPCR was performed with PowerUP SYBR Green Master Mix (Applied Biosystems, Waltham, MA) according to provided protocol on an Applied Biosystems QuantStudio 6 Flex Real-Time PCR system. Primers used are as follows: *rpoA*: 5' rpoA & 3' rpoA; *sspA*: 5' sspA_qPCR & 3' sspA_qPCR; *sspB*: 5' sspB_qPCR & 3' sspB_qPCR; *sleC*: 5'sleC_qPCR & 3'sleC_qPCR; *spoVT*: 5' spoVT_qPCR & 3' spoVT_qPCR; *pdaA*: 5' pdaA_qPCR & 3' pdaA_qPCR; *spoIVA*: 5'spoIVA_qPCR & 3'spoIVA_qPCR; *spoIVB*: 5' spoIVB_qPCR_1 & 3'

spoIVB_qPCR_1; *spoIVB2*: 5' spoIVB2_qPCR_1 & 3' spoIVB2_qPCR_1; *spoIIP*: 5' spoIIP_qPCR_1 & 3' spoIIP_qPCR_1; *dpaA*: 5' dpaA_qPCR & 3' dpaA_qPCR; *spoVAC*: 5' spoVAC_qPCR & 3' spoVAC_qPCR; *spoVAD*: 5' spoVAD_qPCR & 3' spoVAD_qPCR; *spoVAE*: 5' spoVAE_qPCR & 3' spoVAE_qPCR.

Analysis was performed by the ΔΔCT method with comparison to internal control *rpoA* and then mutant strains compared to WT (R20291) [77].

**Luciferase assays.** Overnight cultures were back diluted to $OD_{600} = 0.05$ in BHIS supplemented with thiamphenicol. The cultures were grown for 48 hours. Post incubation, the $OD_{600}$ was recorded and the cultures were used for the Nano-Glo Luciferase assay (Promega, Madison, WI). Briefly, 100 μL of culture was put into a standard Optiplate White bottom 96 well plate. 20 μL of buffer/substrate mixture, prepared as per the kit instructions, was added to the culture. The plate was shaken for 3 minutes before the RLU was determined. The RLU was normalized to the $OD_{600}$ [52,53].

For each trial, 2 technical replicates were measured in different positions in the 96 well plate, due to some variation in measurements based on location within the plates.

## Supporting information

**S1 Fig. EMS mutagenesis strategy.** EMS was added to logarithmically growing cultures and incubated for 3 hours. The resulting cells were spread on sporulation medium (70:30) and incubated for 5 days. Subsequently, the growth was harvested, purified, and then then heat shocked at 65°C for 1 hour. After heat treatment, the samples were plated onto 70:30 and again incubated for 5 days. This enrichment process was repeated 3 times before individual colonies were isolated, phenotypes confirmed, and DNA sent for whole genome re-sequencing. The figure was created with BioRender.com.
(TIF)

**S2 Fig. Spore yield is reduced with expression of *spoIVB2* in EMS strains.** Spore yield of the indicated strain was determined as described in Fig 1. pEV indicates an empty vector. A) Strains isolated during EMS. B) Clean strains containing generated *spoIVB2* alleles. All data represents the average of three independent experiments. Statistical analysis by one way ANOVA with Šídák's multiple comparison test. * P<0.05, ** P<0.01, *** P<0.001, **** P<0.0001. B) **a** P<0.0001 in comparison to *C. difficile* Δ*sspA* Δ*sspB*.
(TIF)

**S3 Fig. qPCR of mutant strains: Part 1.** Strains were grown on sporulation medium for 11 hours before RNA extraction. qPCR was performed using SYBR green. Transcripts for the following genes were determined: A) *sspA*, B) *sspB*, C) *spoIVA*, D) *sleC*, E) *pdaA*, F) *spoVT*. Fold change from R20291 was determined with the ΔΔCT method using *rpoA* transcripts as the internal control. All data represents the average of five independent experiments. Statistical analysis by one way ANOVA with Dunnett's multiple comparison test with the mutant strains compared to wild type. * P<0.05, ** P<0.01, *** P<0.001, **** P<0.0001.
(TIF)

**S4 Fig. qPCR of mutant strains: Part 2.** Strains were grown on sporulation medium for 11 hours before RNA extraction. qPCR was performed using SYBR green. Transcripts for the following genes were determined: A) *spoIVB*, B) *spoIVB2*, C) *spoIIP*. Fold change from R20291 was determined with the ΔΔCT method using *rpoA* transcripts as the internal control. All data represents the average of five independent experiments. Statistical analysis by one way ANOVA with Dunnett's multiple comparison test with the mutant strains compared to wild

type. * P<0.05, ** P<0.01, *** P<0.001, **** P<0.0001.
(TIF)

**S5 Fig. qPCR of mutant strains: Part 3.** Strains were grown on sporulation medium for 11 hours before RNA extraction. qPCR was performed using SYBR green. Transcripts for the following genes were determined: A) *dpaA*, B) *spoVAC*, C) *spoVAD*, D) *spoVAE*. Fold change from R20291 was determined with the ΔΔCT method using *rpoA* transcripts as the internal control. All data represents the average of five independent experiments. Statistical analysis by one way ANOVA with Dunnett's multiple comparison test with the mutant strains compared to wild type. * P<0.05, ** P<0.01, *** P<0.001, **** P<0.0001.
(TIF)

**S1 Table. Primers used in this study.**
(DOCX)

**S2 Table. Strains and plasmids used in this study.**
(DOCX)

**S3 Table. Mutations found within suppressor strains.** The mutations identified in EMS treated suppressor strains are sorted by position within the genome and color-coded based on the isolate containing the mutation.
(XLSX)

**S4 Table. Excel spreadsheet containing, in separate sheets, the underlying numerical data for the figures within this article.**
(XLSX)

## Author Contributions

**Conceptualization:** Hailee N. Nerber, Joseph A. Sorg.

**Data curation:** Hailee N. Nerber.

**Formal analysis:** Hailee N. Nerber, Marko Baloh, Joshua N. Brehm, Joseph A. Sorg.

**Funding acquisition:** Joseph A. Sorg.

**Investigation:** Hailee N. Nerber, Joseph A. Sorg.

**Methodology:** Hailee N. Nerber, Marko Baloh, Joshua N. Brehm, Joseph A. Sorg.

**Project administration:** Joseph A. Sorg.

**Supervision:** Joseph A. Sorg.

**Validation:** Hailee N. Nerber, Marko Baloh.

**Writing – original draft:** Hailee N. Nerber, Joseph A. Sorg.

**Writing – review & editing:** Hailee N. Nerber, Marko Baloh, Joshua N. Brehm, Joseph A. Sorg.

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
