## [Decision Letter · Decision Letter 0]

7 Aug 2024

Dear Dr. Sorg,

Thank you very much for submitting your manuscript "The small acid-soluble proteins of Clostridioides difficile regulate sporulation in a SpoIVB2-dependent manner" for consideration at PLOS Pathogens. As with all papers reviewed by the journal, your manuscript was reviewed by members of the editorial board and by several independent reviewers. The reviewers appreciated the attention to an important topic. Based on the reviews, we are likely to accept this manuscript for publication, providing that you modify the manuscript according to the review recommendations.

Your manuscript has been reviewed by three experts on sporulation and small acid soluble proteins. All three reviewers were very positive about the manuscript. They do not believe any additional experiments are required. They did suggest a few minor modification . These are detailed in their reviews but include, i) inclusion of a figure depicting a model for the current findings, ii) better integration of the manuscript's findings with a recently published Molecular Microbiology paper (reference included in Reviewer 2's comments) and iii) suggestions for a few writing modifications. Please consider these suggestions.

Sincerely,

Bruce A. McClane

Academic Editor

PLOS Pathogens

Michael Otto

Section Editor

PLOS Pathogens

Michael Malim

Editor-in-Chief

PLOS Pathogens

orcid.org/0000-0002-7699-2064

Your manuscript has been reviewed by three experts on sporulation and small acid soluble proteins. All three reviewers were very positive about the manuscript. They do not believe any additional experiments are required. They did suggest a few minor modification . These are detailed in their reviews but include, i) inclusion of a figure depicting a model for the current findings, ii) better integration of the manuscript's findings with a recently published Molecular Microbiology paper (reference included in Reviewer 2's comments) and iii) suggestions for a few writing modifications. Please consider these suggestions.

Reviewer Comments (if any, and for reference):

Reviewer's Responses to Questions

**Part I - Summary**

Reviewer #1: This work appears to provide a major addition to regulation of gene expression during sporulation in Bacillus/Clostridium species. Specifically regulation by SASP binding to DNA . While a possible role like this had been suggested a number of years ago, the authors previous work and this revised manuscript make it a reality. Thus the work is certainly novel and significant. However, two weaknesses in the revised ms are as follows, as well as an observation about one of the authors' findings about the phenotype of their SASP mutants..

1. these is no figure showing their model for SASP/SpoIVB2 to allow readers who are not sporulation junkies to more easily follow the authors' argument based on all their data, and I would urge the authors to add this to allow readers to follow along in the Discussion.

2. While the role the authors role for SASP in activating SpoIVB expression is reasonable in light of all the results, some further proof is needed, and this might involve looking at in vitro RNA transcription from the spoIVB gene (plus upstream/downstream regions) and +/- SASP A and or B, and using RNA polymerase with Sig F or G or A. However, this is certainly beyond the scope of this manuscript.

The authors also noted that the SASP, while important in spore UV resistance, were not important in spore chemical resistance. Given that SASPless spores have little or no cortex, I think that their low chemical resistance is to be expected since cortex-less spores would be expected to have a much more permeable inner membrane (allowing nasty chemicals in!), as well as less CaDPA and thus a higher core water content so reactions in the core would be more rapid. I would also expect that wet heat resistance would also plummet in SASPless spores, and even more than in SASPless B. subtilis or C. perfringens spores.

Reviewer #2: The work unravels a role for the SspA and SspB forespore-specific proteins in the biogenesis of the spore cortex, a function of the mother cell. The work also shows that expression of the sspA/sspB genes is required for proper function of the secretory SpoIVB2 protease, shown before to be required for cortex formation. Together, these observations expand our knowledge of the molecular mechanisms governing spore morphogenesis in C. difficile, in particular how the SspA/B protein may control gene expression and how the forespore and the mother cell cooperate during spore morphogenesis.

Reviewer #3: The revised manuscript by Berber et al. provides compelling new data supporting a revised hypothesis for why the severe sporulation phenotype of a ∆sspA∆sspB deletion or ∆sspB* mutation can be suppressed by a point mutation or silent mutation in the gene encoding SpoIVB2. Importantly, the catalytic activity of SpoIVB2 is important for this suppression, but the mechanism by which these mutations were able to suppress the severe sporulation defect of the sspA and sspB mutants was not clear in the original submission. The authors’ new data strongly suggest that the suppressor mutations increase the translational efficiency or stability of the spoIVB2 transcript. For reasons that remain unclear, increased SpoIVB2 levels allow the ∆sspA∆sspB mutation to be bypassed. The authors propose a model in which SpoIVB2 is able to degrade a factor that likely controls a checkpoint required for sporulation to proceed in the absence of SspA or SspB.

The new data provide a more compelling model for explaining their data, and overall, the revised data provided seem to have less variability than the original submission. The inclusion of the appropriate controls for all the figures adds rigor to the manuscript and improves its readability significantly.

**Part II – Major Issues: Key Experiments Required for Acceptance**

Reviewer #1: Only thing I would emphasize here is including a figure outlining the author's model for SASP modulating sporulation!

Reviewer #2: I reading the responses to all of the reviewer´s comments, I think the authors provided answers to all of them and the manuscript was significantly improved. In particular, the authors revised their hypothesis that the SASPs were repressing expression of spoIVB2; they now show that the SASPs are required for the maintenance of SpoIVB2 levels in the fofrespore.

Reviewer #3: (No Response)

**Part III – Minor Issues: Editorial and Data Presentation Modifications**

Reviewer #1: 1. Further suggest ing a summary figure outlining SASP regulation of sporulation!

2. I fould the meaning of the sentence on l 284/285 somewhat confusing.

3. l 329 - Is this spoVAEb or spoVAEa, or does C. difficile only have Eb??

4. In all of t he strains used in the ms, I imagine some hadplasmids and or antibiotic markers. There have been several papers from Stanley Bruls lab showing that introduction of plasmids.antibiotic markers alters te spore proteome markedly in Bacillus species. Might this be a cause of effects (at least partially) seen in some strains used in the ms?

Reviewer #2: Lines 97-98: I think it can now be stated that the low levels of SpoIVB2 may halt sporulation because cortex synthesis, which depends on SpoIVB2, does not occurs as shown in strain 630Derm (Molecular Microbiology. 2024;00:1–17; this ref is in fact included in the authors list). In the reference above, SpoIVB is referred to as SpoIVB1 and this nomenclature could be maintained for clarity.

Line 144, suggestion: did not form the cortex layer and had visible, but anomalous coat and exosporium layers.

Line 146: by restoring formation of the cortex layer.

Line 191: add ref above, related to the active site of SpoIVB2.

Line 191: the ref above could be added as it discusses the active site of SpoIVB2.

Line 213: to further evaluate this /this should not be crossed)...

Line 242: many problems is a little too vague. Show several deviations from the wt?

Line 246: the coat and the exosporium?

Line 354: not in the present context, but it would be very interesting to examine the phenotype resulting from over expression of spoIVB2!

Line 376: ... under SpoVD and potentially SpoVE control...

Line 463-464: SpoIIP was shown to bed cleaved by SpoIVB2 in the reference above.

Reviewer #3: I have some very minor suggestions for improving readability.

A few very minor points:

In the response to review, the authors state that Fimlaid et al. showed that strain 630 does not complete engulfment. However, in that paper (and in Fimlaid et al. 2015 PLoS Genet), the JIR8094 strain was used rather than strain 630.

Line 93: the statement that the SigG-dependent production of the SASPs leads to the SigF-dependent activation of spoIVB2 is a bit tricky, since at this stage it might be hard to distinguish whether SigF or SigG is up-regulating spoIV2 expression.

Line 186: Consider revising the hypothesis's framing to improve readability. e.g., revising the sentence so that it reads: “From these data, we hypothesize that SspA and SspB indirectly activate the expression of spoIVB2 in WT cells in the late stages of sporulation. Thus, in the ∆sspA∆, sspB, and ∆sspB* mutant backgrounds, SpoIVB2 levels are reduced, resulting in the failure to proteolytically activate a target that is essential for sporulation.” (or degrade a target that inhibits sporulation)

Line 191: consider revising the sentence so that it reads “The suppressor strains potentially increase the amount of SpoIVB2 present, bypassing the need for SigG-dependent induction of spoIVB2 transcription.”

Line 221: please replace “allowed for” with “appeared to allow for”

Line 222: could you add a short descriptor for what is anomalous about the coat?

Line 323: Please add the Shrestha et al. Nat Comm 2023 reference because this is the first description of a spoVE mutant (lacks a cortex also).

Line 373: consider adding “(Fig S4)” to after “statistical significance”

Line 382: I think it would be better to replace “activity” with “levels”

Line 384: consider changing “increased” to “increase”

The authors should also consider adding the Zhou et al. Mol Cell reference (Ben-Yehuda lab), which highlights how SspA and SspB affect the localization of RpoC in dormant spores, which highlights how SspA and SspB can affect transcription within spores (or outgrowing spores).

For Figure 9, please consider using a designation of “PsspA::spoIVB2” for explicitly stating the promoter that is driving spoIVB2. The double colon highlights the origin of the promoter being used.

For Figure 10, please add “of spoIVB2” after “continued expression”

For the middle panel, SpoIVB2 should refer to the gene (lower case start and italicized.)

PLOS authors have the option to publish the peer review history of their article (what does this mean?). If published, this will include your full peer review and any attached files.

Reviewer #1: **Yes: **peter setlow

Reviewer #2: No

Reviewer #3: No

Figure Files:

Data Requirements:

Reproducibility:

References:

---

## [Editor Report · Decision Letter 1]

14 Aug 2024

Dear Dr. Sorg,

We are pleased to inform you that your manuscript 'The small acid-soluble proteins of Clostridioides difficile regulate sporulation in a SpoIVB2-dependent manner' has been provisionally accepted for publication in PLOS Pathogens.

Best regards,

Bruce A. McClane

Academic Editor

PLOS Pathogens

Michael Otto

Section Editor

PLOS Pathogens

Michael Malim

Editor-in-Chief

PLOS Pathogens

orcid.org/0000-0002-7699-2064

The authors have responded adequately to the Reviewer comments. No further revisions needed.
---

## [Editor Report · Acceptance letter]

26 Aug 2024

Dear Dr. Sorg,

We are delighted to inform you that your manuscript, "The small acid-soluble proteins of Clostridioides difficile regulate sporulation in a SpoIVB2-dependent manner," has been formally accepted for publication in PLOS Pathogens.

Best regards,

Michael Malim

Editor-in-Chief

PLOS Pathogens

orcid.org/0000-0002-7699-2064